# LFPT5: A Unified Framework for Lifelong Few-shot Language Learning Based on Prompt Tuning of T5

**Chengwei Qin**♣ **and Shafiq Joty**♣♠

♣ Nanyang Technological University
♠ Salesforce Research
{chengwei003@e.ntu, srjoty@ntu}.edu.sg

## ABSTRACT

Existing approaches to lifelong language learning rely on plenty of labeled data for learning a new task, which is hard to obtain in most real scenarios. Considering that humans can continually learn new tasks from a handful of examples, we expect the models also to be able to generalize well on new few-shot tasks without forgetting the previous ones. In this work, we define this more challenging yet practical problem as Lifelong Few-shot Language Learning (LFLL) and propose a unified framework for it based on prompt tuning (PT) of T5. Our framework called LFPT5 takes full advantage of PT's strong few-shot learning ability, and simultaneously trains the model as a task solver and a data generator. Before learning a new domain of the same task type, LFPT5 generates pseudo (labeled) samples of previously learned domains, and later gets trained on those samples to alleviate forgetting of previous knowledge as it learns the new domain. In addition, a KL divergence loss is minimized to achieve label consistency between the previous and the current model. While adapting to a new task type, LFPT5 includes and tunes additional prompt embeddings for the new task. With extensive experiments, we demonstrate that LFPT5 can be applied to various different types of tasks and significantly outperform previous methods in different LFLL settings.

## 1 INTRODUCTION

A hallmark of human intelligence is that they can learn new tasks quickly by leveraging previously acquired knowledge from other related tasks, and they do so without forgetting prior knowledge. However, despite the monumental success of deep learning in recent years, models face challenges to retain and accumulate knowledge when learning new tasks due to the shift of data distribution – they run into the *overfitting* issue when the data for the new task is small and they forget prior knowledge, a phenomenon known as *catastrophic forgetting* (McCloskey & Cohen, 1989).

Researchers in Lifelong Learning (Thrun & Mitchell, 1995) have proposed a number of methods to alleviate the above issues with machine learning. When it comes to language, earlier approaches to Lifelong Language Learning (LLL) merely focus on a single type of NLP tasks (Wang et al., 2019; d'Autume et al., 2019); see (Biesialska et al., 2020) for a survey. In contrast, humans can easily handle tasks that vary with respect to not only domain but also task type (Figure 1). More recent methods attempt to learn from different types of tasks. These include LAMOL (Sun et al., 2019) and its improvements (Chuang et al., 2020; Sun et al., 2020; Kanwatchara et al., 2021). Despite the effectiveness of these methods in LLL, there are several limitations. First, they all assume plenty of training data for every task which is hard to acquire in most real scenarios as getting large labeled datasets is often expensive and time-consuming. Second, they mainly consider tasks from the decaNLP challenge (McCann et al., 2018) that can be easily framed as question answering (Kumar et al., 2016), paying little attention to sequence labeling tasks such as Name Entity Recognition (NER). Finally, they fine-tune the entire model for all tasks ignoring the possibility of *negative transfer* (Lopez-Paz & Ranzato, 2017) between different types of tasks.

Our work in this paper aims to address these limitations of LLL. We focus on a more challenging yet more practical problem where the model needs to generalize well on new *few-shot* tasks without forgetting the previous ones. We regard this as Lifelong Few-shot Language Learning (LFLL) and investigate three different kinds of tasks: sequence labeling tasks, text classification tasks and text generation tasks.

Based on the strong few-shot learning ability of *prompt tuning* (Lester et al., 2021) of T5 (Raffel et al., 2019), we propose a unified framework for LFLL, named LFPT5 (**L**ifelong **F**ewshot Language Learning with **P**rompt Tuning of **T5**). Specifically, we reframe all types of tasks into a text-to-text format (Figure 2). To continually learn new domains of a task, we simultaneously train the prompt embeddings designed for this task type as a *task solver* and a *data generator* keeping the backbone T5 frozen. When LFPT5 goes about learning a new domain, it first generates pseudo labeled samples of previously learned domains, which are then combined with the new domain training data to alleviate catastrophic forgetting. To achieve *label consistency* between the previous and the current model, LFPT5 also minimizes a KL divergence loss. For the adaptation from one task

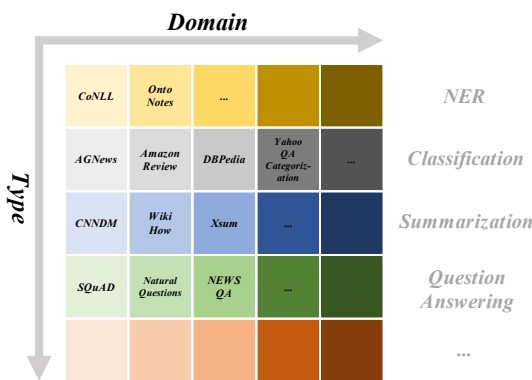

Figure 1: Two different dimensions of lifelong language learning. The horizontal axis (*Domain*) indicates tasks of the same type (*e.g.,* NER), whereas the vertical axis (*Task*) indicates different kinds of tasks.

type to another, LFPT5 includes additional prompt embeddings for the new task, and tunes them similarly. In this way the learning of new tasks minimally affects previously acquired knowledge, mitigating the catastrophic forgetting problem. In the whole learning process, the pre-trained T5 acts as a *meta-learned* model (Brown et al., 2020) that is kept frozen, while the tunable soft prompt acts as a task or domain *adaptation* model. In summary, our main contributions are:

- To the best of our knowledge, we are the first to consider LFLL, a challenging yet practical problem. We propose LFPT5, a unified framework for LFLL based on prompt tuning of T5. LFPT5 can generalize well on various new few-shot tasks without severe forgetting of previously acquired knowledge, which can be seen as an important step towards *general language intelligence*.
- With extensive experiments and analysis, we demonstrate that LFPT5 outperforms previous baselines by a large margin. We have open-sourced our code base at `https://github.com/qcwthu/Lifelong-Fewshot-Language-Learning`.

## 2 RELATED WORK

### 2.1 LIFELONG LEARNING

In lifelong learning (LL), the model is expected to learn sequentially from a stream of tasks with different data distributions. The main problem in LL is *catastrophic forgetting* (McCloskey & Cohen, 1989) – the model forgets previously acquired knowledge after learning a new task. Prior approaches to LL can be divided into three categories. First, *architecture-based* methods dynamically adjust the model architecture to learn new knowledge while preventing the forgetting of previously learned tasks (Chen et al., 2015; Rusu et al., 2016; Mallya et al., 2018). Second, *regularization-based* methods constrain the update of parameters that are important to the learned tasks to retain previous knowledge (Li & Hoiem, 2017; Kirkpatrick et al., 2017; Aljundi et al., 2018). Third, *memory-based* methods keep a number of key samples from previous tasks in memory to alleviate forgetting (Lopez-Paz & Ranzato, 2017; Chaudhry et al., 2018; d'Autume et al., 2019). These methods for LL mostly focus on tasks of the same type (referred as domains in this work). Recently, Sun et al. (2019) proposes LAMOL, a general framework designed for lifelong language learning (LLL), where the model needs to continually learn from different domains as well as different types of NLP tasks.

### 2.2 FEW-SHOT LEARNING

Few-shot learning (FL) aims to learn tasks with a few labeled examples. Due to the scarcity of labeled training data, FL faces the problem of over-fitting. Existing methods to overcome over-

fitting include: (*i*) *model-based* methods that explore how to reduce the hypothesis space of the few-shot task (Triantafillou et al., 2017; Hu et al., 2018), (*ii*) *data-based* methods that try to augment additional data to the few-shot set (Benaim & Wolf, 2018; Gao et al., 2020b), and (*iii*) *algorithm-based* solutions that aim to improve strategies for searching for the best hypothesis. Recently, a new paradigm introducing prompts achieves promising results for few-shot language learning as shown by GPT-3 (Brown et al., 2020), PET (Schick & Schütze, 2020) and LM-BFF (Gao et al., 2020a).

### 2.3 PROMPT-BASED LEARNING

Brown et al. (2020) first show that a GPT-3 frozen model can achieve impressive few-shot results through manually designed prompts that provide a natural language description of the task. Since then many efforts have been made on prompt-based learning (PL). In general, PL modifies the original input, often adding a task-specific template or prompt, which usually contains some unfilled slots to let a pre-trained language model probabilistically generate a textual response, from which the final model output can be derived (Liu et al., 2021b). The ongoing research on PL has explored (*i*) methods of prompt designing, including discrete prompts (Schick & Schütze, 2020; Shin et al., 2020; Tam et al., 2021) and continuous or soft prompts (Li & Liang, 2021; Liu et al., 2021c; Lester et al., 2021), (*ii*) applications of PL (Han et al., 2021; Ben-David et al., 2021; Ding et al., 2021), and analysis of prompt-based learning (Liu et al., 2021a; Le Scao & Rush, 2021; Zhong et al., 2021).

**Summary.** Existing work in lifelong language learning aims to learn from a stream of NLP tasks with plenty of training data, while the research in few-shot learning explores how to generalize well on few-shot tasks. In contrast, we focus on a more challenging yet more practical problem lifelong few-shot language learning (LFLL), where the model is expected to continually learn from a stream of few-shot tasks while avoiding overfitting on the new task and forgetting of previously acquired knowledge. We regard LFLL as an important step towards general language intelligence and propose LFPT5 which takes full advantage of the strong few-shot learning ability of prompt tuning.

## 3 METHODOLOGY

In this section, we first formally define the LFLL problem with the two different adaption dimensions of domains and tasks, and then illustrate how we reframe all types of tasks considered in this work into a text-to-text format in T5. Finally, we present the details of our framework LFPT5.

### 3.1 PROBLEM FORMULATION

As shown in Figure 1, we identify two different dimensions of LFLL: learning of new tasks that are of the same type but potentially of different domains (STDD), and learning of new tasks that are of different types (DT). Specifically, STDD involves learning from a stream of domains $\mathbb{D} = (\mathcal{D}^1, \ldots, \mathcal{D}^n)$ that belong to the same type of few-shot task $\mathcal{T}$, such as NER learning from CoNLL03 (Sang & De Meulder, 2003) and subsequently from OntoNotes (Hovy et al., 2006). Each task domain $\mathcal{D}^k$ has its own training set $S_{\text{train}}^k$, validation set $S_{\text{valid}}^k$, and test set $S_{\text{test}}^k$. After the training on $S_{\text{train}}^k$, the model is expected to perform well on all the $k$ domains that it has learned so far and will be evaluated with the same evaluation metric(s) on the combined test set $\hat{S}_{\text{test}}^k = \cup_{i=1}^k S_{\text{test}}^i$.

Different from STDD, in the DT dimension, the model is expected to continually learn from a sequence of different types of few-shot tasks $\mathbb{T} = (\mathcal{T}^1, \ldots, \mathcal{T}^m)$, such as learning of NER (sequence labeling), then text classification, and subsequently text summarization (generation). After learning of $\mathcal{T}^k$, the model will be evaluated on the test set $S_{\text{test}}^i$ of every learned task $\mathcal{T}^i$ separately for $1 \leq i \leq k$ as the evaluation metrics for different kinds of tasks might be different.

In both dimensions of LFLL, we assume that the validation set $S_{\text{valid}}^k$ has the same size as the few-shot training set $S_{\text{train}}^k$, that is, $|S_{\text{valid}}^k| = |S_{\text{train}}^k|$. The set up of using a few-shot validation set is aligned with the overall goal of generalizing well on new tasks with limited labeled data.

### 3.2 LIFELONG FEW-SHOT LANGUAGE LEARNING WITH PROMPT TUNING OF T5 (LFPT5)

Without loss of generality, let $\mathcal{D}_{\text{task}}$ denote the training dataset for any new few-shot task and $\mathcal{D}_{\text{pre}}$ denote a large-scale pre-training dataset. Our goal is to learn a model $\phi$ for the task. Formally,

$$\arg\max_{\phi} \log p(\phi|\mathcal{D}_{\text{task}}, \mathcal{D}_{\text{pre}}) \approx \arg\max_{\phi} \left[\log p(\phi|\mathcal{D}_{\text{task}}, \theta) + \log p(\theta|\mathcal{D}_{\text{pre}})\right] \quad (1)$$

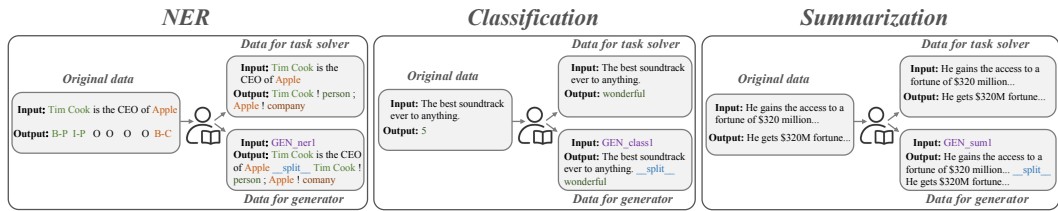

Figure 2: Task formulation for Named Entity Recognition (NER), classification and summarization.

where $\theta$ is a prior pre-trained model, more specifically, a point estimate of the pre-trained model (see A.1). The adaptation task for LFLL thus boils down to solving: $\arg\max_\phi \ \log p(\phi|\mathcal{D}_{\text{task}}, \theta)$. Traditionally, this has been done through fine-tuning $\theta$. However, fine-tuning the entire model effectively on small few-shot tasks could be challenging and may lead to overfitting (Howard & Ruder, 2018).

Brown et al. (2020) show that a large-scale pre-trained model (a frozen GPT-3) can act as a black-box meta-learner (Chen et al., 2017) and yield impressive few-shot performance via manually designed prompts constructed with task descriptions and some canonical examples. As model size continues to increase (often in billions), it is indeed more appealing to have a single generalist model to perform multiple different tasks simultaneously rather than having a separate copy for each task. However, as Lester et al. (2021) pointed out manual prompt engineering may have several key limitations including the human labor involved in the design process which can also be subjective and error-prone, and its rigidness with respect to the maximum sequence length supported by the model. Furthermore, the manual design assumes knowing the task in advance, which limits its applicability to lifelong learning where the next task to learn may not be known in advance.

In our work for LLFL, we adopt the idea of prompt tuning proposed by Lester et al. (2021). We freeze the pre-trained model $\theta$ and prepend a series of tunable tokens $P$, parameterized by $\phi$ (namely, *prompt embeddings*), to the input sequence and optimize $\log p(\phi|\mathcal{D}_{\text{task}}, \theta)$ through gradient descent. We use T5 (Raffel et al., 2019) as the pre-trained meta model, and the prompt embeddings are initialized with the embeddings drawn from the vocabulary of T5.

Prompt tuning is a simple yet effective approach for learning many tasks as it only requires learning a small number of prompt embeddings for each task. In addition, as the prompt embeddings can condense the signal from the training data and exploit the huge amount of meta knowledge contained in the frozen T5 model, prompt tuning also shows impressive results in few-shot learning. These two advantages naturally make prompt tuning a good choice for LFLL.

### 3.2.1 TASK FORMULATION & ADAPTATION

We consider three typical task types in NLP: sequence labeling (*e.g.,* NER), text classification and text generation (*e.g.,* summarization). Inspired by (Raffel et al., 2019; Lester et al., 2021), we re-frame all tasks into a text-to-text format as shown in Figure 2. We denote the input text as $X$ and the output text as $Y$. The training objective for a task with dataset $\mathcal{D}_{\text{task}} = \{(X_1, Y_1), \ldots, (X_n, Y_n)\}$:

$$\mathcal{L}_\phi^{\text{task}} = -\log p(\phi|\mathcal{D}_{\text{task}}, \theta) = -\sum_{i=1}^{n} \log p(Y_i|[P, X_i], \phi, \theta) \tag{2}$$

Where $P$ are the prompt tokens pre-pended to the input and $\phi$ denote their embeddings. Wang et al. (2019) show that memory-based methods where the model preserves some key samples from previous tasks in memory to overcome forgetting, are more effective for lifelong learning in NLP than the other two kinds, architecture and regularization based methods (§2.1). Instead of using an external memory module, we tune our task prompts such that the model simultaneously acts as a task solver and a generator. The generation capability allows the model to generate pseudo samples of previously learned tasks that the current model can use to "refresh" its prior task knowledge.

When training as a **task solver**, the model learns to decode the output text ($Y$) after reading the original input text ($X$). We call this input-output format *TASK format*. For sequence labeling, the output text is split into segment-label pairs by a special token ';', and the text segment and its label in a pair are separated by another special token '!'. For classification, we convert the original label into a natural language description as the output text, *e.g.,* converting the review score 5 into 'wonderful' for sentiment analysis. For text generation, we simply use the target text as the output text.

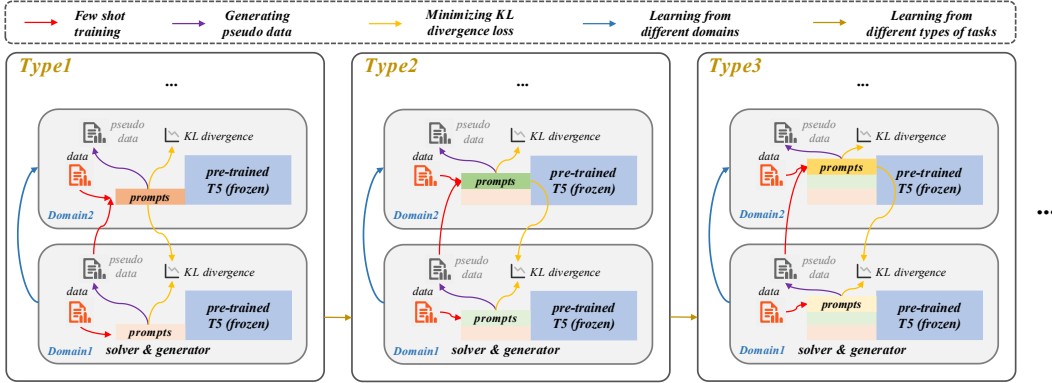

Figure 3: Illustration of the learning process of LFPT5 for different task domains and task types. For learning new domains, LFPT5 simultaneously trains the prompt embeddings as a task solver and a data generator. When a new domain comes, it first generates pseudo samples of previous domains which will be combined with new data for training to mitigate the forgetting of learned knowledge. A KL divergence loss is also optimized to achieve label consistency between the previous and current model. To learn a new task type, LFPT5 includes and tunes additional prompt embeddings for the new task while keeping the previous embeddings frozen.

When training as a **data generator**, the model learns to generate $X$ as well as $Y$ given a task-specific generation token as input; we call this *GEN format*. We use different generation tokens for different types of tasks and different domains to guide the model to generate pseudo samples for a specific task, such as 'GEN_ner1' for CoNLL NER, 'GEN_ner2' for OntoNotes NER and 'GEN_class1' for AGNews classification. In addition, we insert one special token '__split__' between $X$ and $Y$. During inference, the generated pseudo samples which do not contain this special token are discarded. The data generation or language modeling (LM) loss can be expressed as:

$$\mathcal{L}_\phi^{\mathrm{lm}} = -\sum_{i=1}^{n} \log p([X_i, Y_i] | [G, P], \phi, \theta) \tag{3}$$

Where $G$ is a task-specific generation token added to the prompt $P$. The training objective with the *TASK* and *LM* losses becomes: $\mathcal{L}_\phi = \mathcal{L}_\phi^{\mathrm{task}} + \lambda_{\mathrm{lm}} \mathcal{L}_\phi^{\mathrm{lm}}$, where $\lambda_{\mathrm{lm}}$ is the weight of the LM loss. Figure 3 illustrates the complete learning process of LFPT5 for new domains and task types.

**Adapting to New Domains** Before learning on a new domain $\mathcal{D}^k$, LFPT5 first generates pseudo samples $(\tilde{X}, \tilde{Y})$ of previous domains $\mathcal{D}^1, \ldots, \mathcal{D}^{k-1}$ using the corresponding generation token in the input prompt, which will be replayed later to alleviate forgetting of learned knowledge. To achieve *label consistency* on the pseudo samples, we also minimize a KL divergence loss between the previous and current models for the output tokens. More formally,

$$\mathcal{L}_\phi^{\mathrm{KL}} = \sum_{i=1}^{m} \sum_{j=1}^{t} D_{\mathrm{KL}}(p_j(\mathcal{V} | [P, \tilde{X}_i], \phi', \theta) || p_j(\mathcal{V} | [P, \tilde{X}_i], \phi, \theta)) \tag{4}$$

where $m$ is the number of pseudo samples, $t$ is the number of tokens in $\tilde{Y}_i$, $\mathcal{V}$ is the T5 vocabulary and $\phi'$ is the prompt embeddings of the previous model.

The overall loss that LFPT5 optimizes for adapting to new domains is: $\mathcal{L}_\phi = \mathcal{L}_\phi^{\mathrm{task}} + \lambda_{\mathrm{lm}} \mathcal{L}_\phi^{\mathrm{lm}} + \lambda_{\mathrm{kl}} \mathcal{L}_\phi^{\mathrm{KL}}$, where $\lambda_{\mathrm{kl}}$ is the weight of KL divergence loss.

**Adapting to New Task Types** In order to learn a new task type $\mathcal{T}_k$ while not forgetting the acquired knowledge of previous tasks $\mathcal{T}_1, \ldots, \mathcal{T}_{k-1}$, we include an additional set of prompt tokens for the new task and fine-tune their embeddings while keeping the old ones frozen. This is indeed an instance of dynamically expandable network (Yoon et al., 2017), where each task type has its own dedicated prompt. The embedding of the new prompt tokens can be initialized with the learned embeddings from a previous task to avoid forgetting and to have a better task prior. We define 300 tunable tokens per task prompt, meaning that we only add about 0.04% of the parameters of the pretrained T5 when learning a new task type. Compared with previous lifelong learning frameworks which fine-tune the entire model for all tasks ignoring the negative transfer between different types of tasks, LFPT5 shows significant superiority, and it can also achieve better results than multitask learning (§4.4).

# 4 EXPERIMENTS

## 4.1 EXPERIMENT SETUP

**Tasks, Datasets and Metrics** Three different types of tasks are investigated in our work: NER as an instance of sequence labeling, text classification, and summarization as an instance of text generation. For NER, we use CoNLL03 (Sang & De Meulder, 2003) and OntoNotes (Hovy et al., 2006) as different domains. For classification, we conduct experiments on four different datasets/domains: AGNews for news classification (Zhang et al., 2015), Amazon Review for sentiment analysis (McAuley et al., 2015), DBPedia for Wikipedia article classification into topics (Lehmann et al., 2015), and Yahoo for QA categorization (Zhang et al., 2015). The datasets for summarization include CNNDM containing CNN/DM news (Nallapati et al., 2016), WikiHow containing how-to instructions (Koupaee & Wang, 2018) and Xsum containing BBC news (Narayan et al., 2018).

We conduct 16-shot learning for NER and classification based on Gao et al. (2020a), *i.e.,* there are 16 samples per class in the training and validation sets. For summarization, we sample 64 examples for training and validation per domain (see A.9 for details). For pseudo data, LFPT5 generates 2 samples per learned class for NER and classification, and 4 samples per learned domain for summarization. The evaluation metrics of NER, classification and summarization are F1, accuracy and ROUGE scores, respectively. As the task order and few-shot data might influence the performance, we run every experiment 3 times with different random seeds and report the mean results.

**Methods Compared** We use T5-Large as the backbone model and compare our LFPT5 with the following methods in the experiments for learning new domains of a task:

- **Fine-tuning (FT)** tunes the whole T5 model during the LFLL process. We include this method as fine-tuning is still the dominant paradigm in NLP.
- **Prompt tuning (PT)** continually tunes the prompt embeddings while learning on different domains. PT does not include LM and KL objectives and does not generate pseudo samples.
- **EWC** (Kirkpatrick et al., 2017) and **MAS** (Aljundi et al., 2018) are two regularization-based lifelong learning methods requiring no extra memory. They constrain the update of parameters that are important to the learned tasks to retain previous knowledge. We apply these two methods to both PT and FT, and get four distinct methods: **EWC-PT**, **MAS-PT**, **EWC-FT** and **MAS-FT**.
- **Prompt tuning with real data (PT-R)** selects the same number of randomly selected *real* samples from the learned domains as the generated pseudo samples in LFPT5. These samples are used as memory data which is replayed during the learning of the new domain. PT-R resembles a 'real' memory-based LFLL model with prompt tuning and its performance can be used to compare the quality of the pseudo samples generated by LFPT5.
- **Multitask prompt tuning (MT-PT)** simultaneously trains on all the domains together with the combined data. It serves as an *upper bound* for LFPT5 which can use only the new domain data.

In addition, we report experiments with a different backbone model (T5-Base), different numbers of few-shot data and different number of pseudo samples in Appendix A.4, A.5 and A.6, respectively.

For adapting to new task types, we compare LFPT5 with multitask fine-tuning (MT-FT), MT-PT and **AdapterFusion** (Pfeiffer et al., 2021) which learns a task-specific composition of adapters from previous tasks.

## 4.2 SINGLE TASK RESULTS

To assess the learning ability of prompt tuning, we first compare single task few-shot results for T5 fine-tuning (T5-FT), T5 prompt tuning (T5-PT) and BERT-Large fine-tuning on NER and classifica-

| Method | NER | | Text classification | | | |
|---|---|---|---|---|---|---|
| | CoNLL03 | OntoNotes | AGNews | Amazon | DBPedia | Yahoo |
| **SoTA (full-shot)** | 94.6 | 92.07 | 95.55 | 65.83 | 99.38 | 77.62 |
| **BERT-Large** | $62.67_{\pm 1.34}$ | $\mathbf{63.55}_{\pm 1.68}$ | $82.33_{\pm 1.66}$ | $40.47_{\pm 1.39}$ | $97.29_{\pm 0.61}$ | $59.97_{\pm 2.25}$ |
| **T5-FT** | $53.74_{\pm 1.20}$ | $55.15_{\pm 0.70}$ | $83.17_{\pm 2.60}$ | $\mathbf{48.80}_{\pm 2.05}$ | $\mathbf{98.19}_{\pm 0.19}$ | $50.07_{\pm 21.84}$ |
| **T5-PT** | $\mathbf{68.40}_{\pm 1.24}$ | $61.23_{\pm 2.14}$ | $\mathbf{85.33}_{\pm 1.05}$ | $43.73_{\pm 0.41}$ | $97.36_{\pm 0.52}$ | $\mathbf{65.67}_{\pm 2.03}$ |

Table 1: Results on single few-shot tasks on NER (F1 score) and text classification (accuracy).

tion in Table 1, while Figure 4 shows the comparison between T5-FT and T5-PT on summarization. We also report the state-of-the-art (SoTA) results for the original full-shot training for each task.

We can see that the performance of T5-PT is quite good compared with BERT-Large and T5-FT. T5-FT overfits on several few-shot tasks (CoNLL03, OntoNotes and Yahoo) and achieves poor results. PT significantly improves these results as it requires to tune only the prompt embeddings. In particular, T5-PT achieves better results than fine-tuned BERT-Large in all cases except OntoNotes NER. Similarly, on summarization, T5-PT achieves better performance than T5-FT in all measures across the datasets except ROUGE-1 on WikiHow. These results suggest that PT has the potential for LFLL if we can solve the catastrophic forgetting problem well.

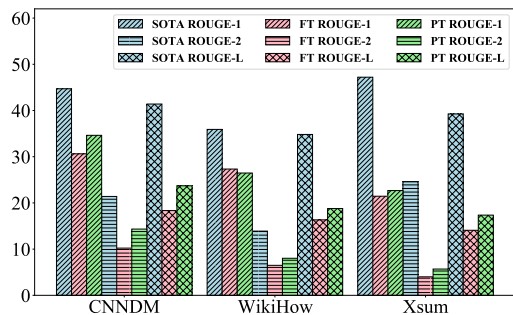

Figure 4: Results for T5 prompt tuning (PT) and T5 fine-tuning (FT) on summarization (ROUGE scores).

### 4.3 RESULTS FOR LEARNING NEW DOMAINS

**NER** The LFLL results on the NER domains are shown in Table 2. We report the final F1 score on the whole test set after learning all domains. We observe that EWC and MAS achieve slightly better results than simply fine-tuning the parameters, meaning the catastrophic forgetting problem is still severe. LFPT5 outperforms these two regularization-based lifelong learning methods by a large margin, which demonstrates the superiority of our method.

| Method | Fine-tuning | EWC- Fine-tuning | MAS-Fine-tuning | Prompt tuning-Real | MT-Prompt tuning |
|--------|-------------|------------------|------------------|---------------------|-------------------|
| **F1** | $43.07_{\pm1.48}$ | $43.53_{\pm1.7}$ | $43.63_{\pm1.9}$ | $48.72_{\pm0.9}$ | $\mathbf{54.32_{\pm0.88}}$ |
| **Method** | | Prompt tuning | EWC-Prompt tuning | MAS-Prompt tuning | LFPT5 |
| **F1** | | $44.34_{\pm0.46}$ | $44.68_{\pm1.4}$ | $45.09_{1\pm.45}$ | $\mathbf{47.59_{\pm2.16}}$ |

Table 2: F1 score on the whole test set after learning all NER domains (CoNLL03, OntoNotes).

Comparing the results of PT- and FT-based methods, we can find that PT-based methods show better performance, which can be interpreted by two factors: (*i*) PT has stronger ability than FT for few-shot learning of new domains. (*ii*) The knowledge of the two domains is not so difficult to transfer from one to the other as there are some overlaps between the label spaces. So even if PT needs to continually learn knowledge from different domains with much fewer tunable parameters than FT, it can successfully do so and outperform FT. PT-R performs better than LFPT5, which means that the quality of generated pseudo samples could be further improved. In addition, there is a performance gap between LFPT5 and MT-PT, indicating there still remains room for improvement.

| Method | Fine-tuning | EWC- Fine-tuning | MAS-Fine-tuning | Prompt tuning-Real | MT-Prompt tuning |
|--------|-------------|------------------|------------------|---------------------|-------------------|
| **Accuracy** | $40.11_{\pm7.76}$ | $40.60_{\pm3.02}$ | $40.79_{\pm6.09}$ | $67.23_{\pm1.36}$ | $\mathbf{76.08_{\pm0.77}}$ |
| **Method** | | Prompt tuning | EWC-Prompt tuning | MAS-Prompt tuning | LFPT5 |
| **Accuracy** | | $28.47_{\pm9.65}$ | $29.09_{\pm8.92}$ | $29.46_{\pm8.97}$ | $\mathbf{52.71_{\pm4.19}}$ |

Table 3: Accuracy on the whole test set after learning all domains (AGNews, Amazon, DBPedia, Yahoo).

**Text Classification** Table 3 shows the classification results on the whole test set after learning the four domains. We can see that LFPT5 achieves significant improvements compared with previous lifelong learning methods. For text classification, a significant difference from NER is that FT-based methods show much better performance than PT-based methods. We analyse the reasons as follows. The label space of the four domains is quite different, which makes it hard to transfer knowledge across different domains. So retaining and accumulating knowledge during the learning of different domains is pretty challenging for the PT-based methods as they have only a few tunable parameters. Acquiring of new information can easily cause forgetting of previously learned knowledge. Compared with PT, there are much more tunable parameters in FT, improving its ability to accommodate knowledge from different domains. Even though LFPT5 is based on PT, it can overcome such limitations by learning to remember consistently from its own generated pseudo samples.

In Appendix A.3, we additionally evaluate how LFPT5 performs compared to the baselines for a large number of different tasks (domains) by considering 5 NLI tasks and combine them with the

original 4 classification tasks to form a longer sequence of 9 classification tasks. These results verify that LFPT5 performs much better than previous baselines when learning from many tasks.

| Method | Fine-tuning | EWC- Fine-tuning | MAS-Fine-tuning | Prompt tuning-Real | MT-Prompt tuning |
|---|---|---|---|---|---|
| **A-RG** | $15.71_{\pm 1.35}$ | $15.91_{\pm 1.46}$ | $15.76_{\pm 1.71}$ | $17.48_{\pm 0.25}$ | $\mathbf{19.78}_{\pm 0.70}$ |
| **Method** | | Prompt tuning | EWC-Prompt tuning | MAS-Prompt tuning | LFPT5 |
| **A-RG** | | $15.67_{\pm 0.24}$ | $15.85_{\pm 0.15}$ | $15.79_{\pm 0.09}$ | $\mathbf{17.05}_{\pm 0.92}$ |

Table 4: Average of ROUGE-1, ROUGE-2 and ROUGE-L scores (A-RG) on the whole test set after learning all domains (CNNDM, WikiHow, XSum).

**Summarization**    For summarization, we find that the generated pseudo summaries (that follow the generated pseudo source documents) are often ambiguous. This could be because summarization has a large search space and is often an underconstrained task for the model as showed by Kryscinski et al. (2019). As the leading three sentences (*a.k.a.* Lead-3) already construct a strong baseline for summarization (especially for news articles), we use the leading three sentences of the generated document as its summary to form the pseudo data. From the results in Table 4, we can see that PT-based methods achieve similar performance to FT-based methods. This is different from NER and text classification, showing that the difficulty of transferring knowledge across different domains in summarization might be between that of NER and classification. Here also LFPT5 outperforms previous lifelong learning methods by a large margin.

**Summary** LFPT5 achieves much better performance than previous lifelong learning methods on three different types of tasks, which verifies its effectiveness and strong generalization ability.

## 4.4 Results for Learning New Task Types

To investigate LFPT5's performance on learning new task types, we consider two different variants: (*i*) **LFPT5 with FKT** initializes the prompt embeddings of one task using the prompt embeddings of the previously learned task, which we regard as forward knowledge transfer (FKT), and (*ii*) **LFPT5 w.o. FKT** initializes the prompt embeddings of every task with the embeddings drawn from the vocabulary of T5. For these experiments, we use CoNLL03 for NER, AGNews for text classification and CNNDM for summarization. From the results in Table 5, we can observe the following:

- Both variants of LFPT5 can achieve better performance than MT-FT and MT-PT. Multitask learning simultaneously trains all tasks together. The learning of one task might cause negative effect on the learning of others. In contrast, LFPT5 variants include and tune additional prompt embeddings for new types of tasks which avoids the negative cross-task knowledge transfer.

- LFPT5 performs better than AdapterFusion (Pfeiffer et al., 2021) which demonstrates its superiority. Moreover, LFPT5 is much more parameter-efficient than AdapterFusion. In the course of learning these three different task types, AdapterFusion introduces about 21.72% of the parameters of the pretrained T5, while LFPT5 only adds about 0.12%.

- Comparing the two variants of LFPT5, the effect of forward knowledge transfer can be positive or negative, depending on the tasks. The forward knowledge transfer between classification and summarization is positive. However, they have negative effect on NER; transferring knowledge from them to NER or from NER to them negatively affect the learning of the new task.

| Method | Task Order | | |
|---|---|---|---|
| | (i)
Summ-Class-NER | (ii)
Class-NER-Summ | (iii)
NER-Summ-Class |
| Multitask fine-tuning | 23.24, 78.25, 57.81 | 81.50, 58.28, 21.28 | 50.21, 22.49, 82.25 |
| Multitask prompt tuning | 24.16, 85.50, 50.80 | 82.75, 65.31, 23.36 | 62.83, 11.51, 83.25 |
| AdapterFusion | 22.26, 83.25, 55.62 | 81.25, 63.19, 22.37 | 62.99, 21.20, 82.50 |
| LFPT5 w.o. FKT | **25.48**, 84.75, **63.28** | **83.25**, 67.66, 23.68 | **66.65**, **22.97**, **84.50** |
| LFPT5 with FKT | **25.48**, **86.00**, 62.44 | **83.25**, 65.01, **24.92** | **66.65**, 22.80, 84.25 |

Table 5: Results for learning three different task types: NER (CoNLL), Classification (AGNews) and Summarization (CNNDM). The tasks are presented in three different orders with different few-shot samples (results are shown in the same order). The metrics reported are F1 for NER, accuracy for Classification and Average-ROUGE for Summarization.

| Method | Domain Order | | | Average |
|---|---|---|---|---|
| | (i)
DB-Amazon-Yahoo-AG (Avg.) | (ii)
DB-Amazon-AG-Yahoo (Avg.) | (iii)
Yahoo-Amazon-AG-DB (Avg.) | |
| Prompt tuning | 00.00, 00.00, 07.29, 81.71 (18.88) | 00.00, 00.00, 52.57, 64.57 (24.85) | 00.71, 00.00, 00.00, 97.86 (41.67) | $28.47_{\pm 9.65}$ |
| EWC-Prompt tuning | 00.00, 00.00, 10.86, 82.43 (19.79) | 00.00, 00.00, 56.14, 68.14 (26.36) | 00.00, 00.00, 00.00, 96.93 (41.12) | $29.09_{\pm 8.92}$ |
| MAS-Prompt tuning | 00.00, 00.00, 12.57, 83.86 (20.45) | 00.00, 00.00, 58.00, 65.71 (26.24) | 00.29, 00.00, 00.57, 97.86 (41.70) | $29.46_{\pm 8.97}$ |
| Fine-tuning | 26.71, 06.40, 09.43, 85.71 (32.48) | 21.36, 05.40, 57.00, 71.29 (37.09) | 04.43, 17.80, 25.86, 98.14 (50.76) | $40.11_{\pm 7.76}$ |
| EWC-Fine-tuning | 38.21, 00.20, 21.00, 86.29 (39.00) | 24.07, 04.00, 59.86, 68.14 (37.97) | 00.14, 12.80, 05.86, 98.07 (44.82) | $40.60_{\pm 3.02}$ |
| MAS-Fine-tuning | 37.29, 00.60, 15.71, 83.29 (36.91) | 20.79, 00.40, 61.14, 67.00 (36.06) | 01.00, 11.40, 27.57, 98.07 (49.39) | $40.79_{\pm 6.09}$ |
| Prompt tuning-Real | 85.79, 31.20, 43.14, 82.00 (67.67) | 82.57, 37.40, 67.29, 64.43 (68.64) | 43.71, 24.20, 58.71, 94.29 (65.39) | $67.23_{\pm 1.36}$ |
| LFPT5 | 48.57, 23.20, 32.43, 78.43 (**47.64**) | 54.93, 12.20, 61.86, 67.43 (**52.58**) | 10.57, 09.20, 59.86, 98.00 (**57.91**) | $\mathbf{52.71}_{\pm 4.19}$ |
| MT-Prompt tuning | 95.00, 47.40, 62.29, 75.57 (**76.73**) | 93.57, 47.80, 73.86, 65.57 (**76.52**) | 61.71, 43.00, 74.71, 93.21 (**75.00**) | $\mathbf{76.08}_{\pm 0.77}$ |

Table 6: Text classification accuracy on the whole test set for three runs with different domain order.

## 5 ANALYSIS

**Influence of Domain Order**   To evaluate the influence of domain orders when LFPT5 is learning different task domains, we show the results of three runs with different domain order on the classification task in Table 6. We can see that the order of domains influences the performance of all methods a lot. For example, PT can achieve 41.67 accuracy on the third run while the accuracy of the first run is only 18.88. This phenomenon indicates that the difficulty of transferring knowledge from one domain to another might be quite different from that of the opposite transfer direction. Though the performance is affected by the order, LFPT5 outperforms previous regularization-based lifelong learning methods by a large margin for all different orders (see Appendix A.8 for more analysis).

| $\lambda_{kl}$ | 0 | 0.02 | 0.04 | 0.10 | 0.20 | 0.40 |
|---|---|---|---|---|---|---|
| **A-RG** | $16.27_{\pm 0.50}$ | $16.41_{\pm 0.27}$ | $17.05_{\pm 0.92}$ | $\mathbf{17.11}_{\pm 0.59}$ | $16.97_{\pm 0.88}$ | $16.26_{\pm 0.32}$ |

Table 7: Average Rouge (A-RG) score of LFPT5 with different $\lambda_{kl}$ on summarization.

**Influence of KL Loss**   To investigate the influence of the label consistency loss $\mathcal{L}^{\text{KL}}$ (Eq. 4), we conduct experiments with different $\lambda_{kl}$ on summarization. From the results in Table 7, we observe that the model achieves the best A-RG score of 17.11 with $\lambda_{kl} = 0.10$ and score of 16.26 with $\lambda_{kl} = 0.40$. The performance of the variant without $\mathcal{L}^{\text{KL}}$ (*i.e.*, $\lambda_{kl} = 0$) is worse than the performance of all other variants except the variant with $\lambda_{kl} = 0.40$ (too large), which demonstrates the effectiveness of $\mathcal{L}^{\text{KL}}$.

**Quality of Pseudo Samples**   We show a few pseudo samples generated by LFPT5 in Figure 5. We can observe that LFPT5 can generate high-quality pseudo samples which are useful for remembering previous knowledge. However, as shown in the right part of the figure, the label of generated data could also be incorrect, which explains the performance gap between LFPT5 and PT-R. In addition, there are several obvious errors, *e.g.,* the pseudo data might not have the '__split__' token or belong to the re-

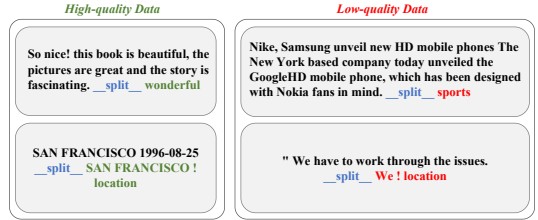

Figure 5: Examples of generated pseudo samples for text classification (top) and NER (bottom).

quired domain. We can automatically discard these samples. We believe that exploring methods to generate more reliable pseudo data should be a quite promising research direction in LFLL.

**Abbreviation Variations**   When learning NER, LFPT5 as a task solver needs to generate the entities in the original input (Figure 2). We observe an entity error related to abbreviation during the generation, such as generating 'the United States' while the original entity is 'U.S.'. This kind of error unfairly penalizes LFPT5's F1 score, but it also indicates that T5 does not just copy words from the original input but thinks about the relevant knowledge and expresses it in its own way.

## 6 CONCLUSION

In this work, we introduce LFPT5, a unified framework for lifelong few-shot language learning (LFLL) where the model needs to generalize well on various new few-shot tasks without forgetting previous acquired knowledge. Extensive experimental results and analysis show that LFPT5 can easily adapt to new types of tasks or new domains while retaining the knowledge of learned tasks, which we regard as an important step towards general language intelligence. In the future, we would like to investigate ways to improve the quality of generated pseudo samples.

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

## A  APPENDIX

### A.1  DERIVATION OF EQUATION 1

Assuming $\phi \perp\!\!\!\perp \mathcal{D}_{\text{pre}}|\theta$, we can write:

$$\arg\max_{\phi} \log p(\phi|\mathcal{D}_{\text{task}}, \mathcal{D}_{\text{pre}}) = \arg\max_{\phi} \int_{\theta} [\log p(\phi|\mathcal{D}_{\text{task}}, \theta) + \log p(\theta|\mathcal{D}_{\text{pre}})]d\theta$$

$$\approx \arg\max_{\phi} [\log p(\phi|\mathcal{D}_{\text{task}}, \hat{\theta}) + \log p(\hat{\theta}|\mathcal{D}_{\text{pre}})] \quad \Rightarrow \text{with point estimate of } \theta.$$

### A.2  DERIVATION OF EQUATION 2

$$\mathcal{L}_{\phi} = -\log p(\phi|\mathcal{D}_{\text{task}}, \theta) = -\log[p(\mathcal{D}_{\text{task}}|\phi, \theta)\, p(\phi)]$$

$$= -\sum_{i=1}^{n} \log p(Y_i|[P, X_i], \phi, \theta) \quad \Rightarrow \text{assuming uniform } p(\phi).$$

### A.3  LEARNING FROM A LARGE NUMBER OF DIFFERENT DOMAINS

To evaluate whether our method can perform better than the baselines when learning from a large number of different domains, we consider 5 NLI tasks (GLUE-MNLI (Williams et al., 2017), Scitail (Khot et al., 2018), SICK (Marelli et al., 2014), SuperGLUE-CB (De Marneffe et al., 2019) and GLUE-RTE (Wang et al., 2018)) as classification and combine them with the original 4 classification tasks to form a long sequence of 9 classification tasks. We evaluate LFPT5, MAS-Prompt tuning and MAS-Fine-tuning on this long sequence. The accuracy after learning all tasks is shown in Table 8. From the results, we can observe that LFPT5 still performs much better than previous baselines when learning from a large number of tasks.

| Method | LFPT5 | MAS-Prompt tuning | MAS-Fine-tuning |
|---|---|---|---|
| **Accuray (%)** | **43.98**$_{\pm 2.68}$ | 9.37$_{\pm 3.17}$ | 34.37$_{\pm 6.21}$ |

Table 8: Accuracy (%) of different methods after learning all 9 domains.

## A.4  DIFFERENT BACKBONE MODEL

To investigate how the model scale affects the LFLL capability, we compare the performance of LFPT5, EWC-Prompt tuning and EWC-Fine-tuning on summarization using T5-Base backbone. From the results in Table 9, we can observe that LFPT5 performs much better than EWC-Prompt tuning. However, it is slightly worse than EWC-Fine-tuning. This is consistent with the finding in Lester et al. (2021) that prompt tuning performs better when applied to larger pretrained language models.

| Method | LFPT5 | EWC-Prompt tuning | EWC-Fine-tuning |
|---|---|---|---|
| **A-RG** | 14.93$_{\pm 0.82}$ | 13.05$_{\pm 1.23}$ | **15.15**$_{\pm 1.38}$ |

Table 9: A-RG score of different methods with T5-Base backbone on summarization.

## A.5  DIFFERENT NUMBERS OF FEW-SHOT DATA

We conduct experiments to compare the performance of LFPT5, EWC-Prompt tuning and EWC-Fine tuning with different numbers (16, 32) of few-shot data on summarization. The A-RG score is shown in Table 10. From the results, we can see that LFPT5 consistently outperforms previous baselines with different numbers of few-shot samples.

| Few-shot Number | LFPT5 | EWC-Prompt tuning | EWC-Fine-tuning |
|---|---|---|---|
| 16 | **15.11**$_{\pm 0.44}$ | 14.16$_{\pm 0.42}$ | 13.75$_{\pm 2.35}$ |
| 32 | **15.58**$_{\pm 0.27}$ | 14.30$_{\pm 0.48}$ | 14.78$_{\pm 1.49}$ |

Table 10: A-RG score of different methods with different numbers (**16**, **32**) of few-shot data on summarization.

## A.6  INFLUENCE OF THE NUMBER OF PSEUDO SAMPLES

As generating pseudo samples is feasible and cheaper, we can use any number of pseudo samples. We conduct experiments on summarization to analyze the influence of different numbers of pseudo samples. From the results in Table 11, we can find that increasing the number of pseudo samples will not always improve the performance. The model achieves the best A-RG score 17.05 with 4 pseudo samples.

## A.7  MULTIPLE PROMPTS IN MULTITASK PROMPT TUNING

We use a single prompt for multitask prompt tuning in Table 5, which is different from LFPT5 in model capacity. To better support our claim, we conduct multitask prompt tuning experiments using the same number of tunable tokens as LFPT5 (multiple prompts). The tunable tokens are shared among all three tasks. From the results in Table 12, we can see that LFPT5 performs better than both multitask prompt tuning methods.

## A.8  ANALYSIS ON PT-BASED EWC AND MAS

From the results in Table 6, we can observe that the performance on previous tasks for PT-based EWC and MAS is almost 0. There could be two reasons:

- There are only a few tunable parameters in prompt tuning, which is difficult for retaining and accumulating knowledge. So the learning of new knowledge from different domains is more

| Number | 2 | 4 | 8 | 16 | 32 |
|---|---|---|---|---|---|
| **A-RG** | $16.28_{\pm 0.95}$ | $\mathbf{17.05}_{\pm 0.92}$ | $16.80_{\pm 0.52}$ | $16.74_{\pm 0.45}$ | $16.79_{\pm 0.64}$ |

Table 11: A-RG score of LFPT5 with different numbers of pseudo samples on summarization.

| Method | Task Order | | |
|---|---|---|---|
| | (i) Summ-Class-NER | (ii) Class-NER-Summ | (iii) NER-Summ-Class |
| Multitask prompt tuning (single prompt) | 24.16, 85.50, 50.80 | 82.75, 65.31, 23.36 | 62.83, 11.51, 83.25 |
| Multitask prompt tuning (multiple prompts) | 21.01, 82.25, 58.59 | 83.00, 64.83, 22.73 | 63.23, 21.41, 84.00 |
| LFPT5 w.o. FKT | **25.48**, 84.75, **63.28** | **83.25**, **67.66**, 23.68 | **66.65**, **22.97**, **84.50** |
| LFPT5 with FKT | **25.48**, **86.00**, 62.44 | **83.25**, 65.01, **24.92** | **66.65**, 22.80, 84.25 |

Table 12: Comparison results of single prompt multitask prompt tuning, multiple prompts multitask prompt tuning and LFPT5.

likely to cause the forgetting of previously learned knowledge. LFPT5 utilizes pseudo samples to alleviate this problem.

- *Few-shot* language learning is more challenging. The model training is already sub-optimal even without lifelong learning. So the performance is relatively low.

## A.9 DATASETS DETAILS

There are 4 and 18 classes in CoNLL03 and OntoNotes, respectively. And the number of classes in AGNews, Amazon, DBPedia and Yahoo is 4, 5, 14 and 10, respectively. We sample 16 examples per class, which means that there are *at least* 64 samples in the training and validation sets. Therefore, we sample 64 examples for the training and validation set per domain (dataset) for summarization.

## A.10 PARAMETER SETTINGS

We use the Adafactor (Shazeer & Stern, 2018) optimizer with a learning rate of 0.5. For NER, we set $\lambda_{lm}$ and $\lambda_{kl}$ to 0.10 and 0.03, respectively. For text classification, we adopt 0.25 and 0.01 for the loss weights $\lambda_{lm}$ and $\lambda_{kl}$, respectively. We set 0.10 for $\lambda_{lm}$ and 0.04 for $\lambda_{kl}$ when learning summarization. Hyperparameter search is done on the validation sets when comparing the single task few-shot results in Section 4.2.

For EWC and MAS based methods, we conduct hyper-parameter search and report the optimal results. For AdapterFusion (Pfeiffer et al., 2021), we adopt the implementation from AdapterHub and use the default adapter settings for T5. The default bottleneck reduction factor is 16, *i.e.,* the bottleneck size is 64. We adopt a learning rate of 1e-4 with AdamW and a linear learning rate decay following the orginal AdapterFusion paper. All other hyper-parameter settings (such as batch size and evaluation interval) are the same as LFPT5.

