# OpenReview forum: "LFPT5: A Unified Framework for Lifelong Few-shot Language Learning Based on Prompt Tuning of T5"
_ICLR.cc/2022/Conference — ICLR 2022 Poster_

### Official Review · Reviewer_Ap2b · 2021-10-16

**Correctness:** 2
**Technical Novelty And Significance:** 3
**Empirical Novelty And Significance:** 3
**Recommendation:** 5
**Confidence:** 2

**Main Review:**


Strengths
- Few-shot setting and lifelong learning are important settings.
- I can't give any more strengths at this time because I had a hard time understanding the paper.

Weaknesses (reasons to reject)
- The main contribution of saying that LFPT5 "can be seen as a vital step towards general language intelligence" is surely an overclaim, right? If I ask 100 researchers, how many of them will see this paper as a vital step towards general language intelligence?
- I did not understand why the proposed method is restricted to the few-shot setting. Do gains go away in the full data setting?
- The experiments are OK but not probably not good enough to support the general claims about lifelong learning. E.g., it would be great to show more than 3 tasks.
- I had a hard time understanding the paper, see below (and I will remove this comment from reasons to reject once it is clarified):
	- The key part that I did not understand is why prompt tuning isn't already a sufficient life-long learning method. My understanding of prompt tuning is that no model weights are modified, only the prompt is optimized, and optimized prompts are stored for each task. Then given a new task, a new prompt can be trained and stored. This works because the size of prompts is many orders of magnitude smaller than the size of the model. Maybe another way of asking this question is, how is the following different from regular prompt tuning: "while adapting to a new task type, LFPT5 includes and tunes additional prompt embeddings for the new task"?
	- Another part I do not understand is why pseudo samples need to be generated. Why can't you just use the prior examples?
	- I am missing why this idea of retraining on previous domains each time is a good one. Let's say we're up to 1000 tasks. How much weight does the new task get in this mixture? Won't this take a long time?

Weaknesses (not reasons to reject)
- The introduction could make it more clear why the second two limitations of LLL are indeed limitations. Why NER in particular, instead of other tasks that haven't been explored thoroughly? The negative transfer point needs some more evidence as well, as I believe the decaNLP challenge showed positive transfer, right?

Minor comments
- Is the finetuned BERT-Large a few-shot finetuned or with the full dataset?
- PT is not defined at first use in the abstract
- All the acronyms are pretty confusing
- Typo: review socre 5

**Summary Of The Paper:**

This paper explores lifelong few-shot language learning. Their framework trains the model as a task solver and data generator. They use pseudo data for new domains, and additional prompt embeddings for new tasks.

**Summary Of The Review:**

This paper tackles an important task. I didn't understand why their method specifically applies to the few-shot scenario. I also struggled to understand why the proposed method solves lifelong learning over regular prompt tuning. One sentence in the paper seems like an overclaim to me.

---

> ### Author Response · Authors · 2021-11-19
> **Response to Reviewer Ap2b (Part 1)**
>
> ## Reviewer Ap2b
>
>
> Thanks for your review. We believe there is a gross misunderstanding about the motivation and significance of LFLL and the design details of our work. Please allow use to first explain these points, and then address your concerns. We would appreciate your quick response so that we can address further concerns (if any) within the rebuttal time period.
>
> **Motivation \& significance**
>
> As we describe in Section 1, humans can learn new tasks with only few examples quickly by leveraging previously acquired knowledge from other related tasks, and they do so without forgetting prior knowledge. This is a hallmark of human intelligence. However, it has been very challenging for a model to handle new few-shot tasks (**overfitting**) and remember previous knowledge (**forgetting**). We believe that with the rise of large-scale pre-trained language models which already contain huge amount of meta knowledge, the community should pay more attention to this challenging yet practical promlem: how to efficiently adapt pre-trained language models to new types of tasks and new domains (same type) with only few samples while retaining the knowledge of previously learned tasks.
>
>
> If the above problem is successfully solved, we can say that we realize general language intelligence to some extent. The model serves as a general solver (handling various tasks) and behaves similarly to humans (learning new tasks with only few samples quickly by leveraging acquired knowledge while retaining prior knowledge). Our work takes the first and vital step towards this goal: we formally define this problem as lifelong few-shot language learning (LFLL) and provide a SoTA model and benchmark for future research in this direction.
>
>
> **Our design details**
>
>
> As the prompt embeddings can condense the signal from the training data and exploit the huge amount of meta knowledge contained in the frozen T5 model, prompt tuning shows impressive results in few-shot learning. Our work takes full advantage of the strong few-shot learning ability of prompt tuning to realize LFLL.
>
>
> We categorize two different dimensions in LFLL: (1) learning from new task types, and (2) learning from new domains (same task type). We argue that when the model is learning different domains of the same type (such as CoNLL and OntoNotes for NER), it should use the same shared prompt embeddings. Taking NER as an example, we only provide the model several input sentences without additional information and ask the model to recognize entities in these sentences during inference. Due to the lack of prior domain information, the model only knows that it should serve as a general NER task solver (domain-agnostic). Therefore, the same prompt embeddings should be shared among different domains of NER. In other words, the prompt embeddings are not domain-specific, but type-specific.
>
>
> In the course of learning the new domains of a task type, the prompt embeddings are constantly updated with new task information, thus getting into the risk of forgetting previous domain knowledge. To alleviate the forgetting problem, LFPT5 simultaneously trains the model as a task solver and a data generator. The model can generate pseudo samples of previously learned domains to mitigate forgetting. In addition, a KL divergence loss is introduced to achieve label consistency between the previous (teacher) and the current (student) model so that the current model can retain more learned knowledge.
>
>
> For learning of new types of tasks, we include a different set of prompt tokens for the new task and fine-tune their embeddings while keeping the old ones (not used for the current task) frozen. The new prompt tokens are initialized from the vocabulary of T5 (same as regular prompt tuning) or the prompt tokens of the previously learned task (different from regular prompt tuning) as described in Section 4.4. This method is parameter-efficient as we only introduce a small number of parameters (about 0.04% of the parameters of the pretrained T5) for a new type.
>
>
> ***The main contribution of saying that LFPT5 "can be seen as a vital step towards general language intelligence" is surely an overclaim, right? If I ask 100 researchers, how many of them will see this paper as a vital step towards general language intelligence? I did not understand why the proposed method is restricted to the few-shot setting. Do gains go away in the full data setting?***
>
> 1. We argue that this is not an overclaim; please see **Motivation \& significance** above.
> 2. We do not focus on full data setting as this is not how humans learn. In addition, assuming plenty of training data for every task is hard to acquire in most real scenarios as getting large labeled datasets is often expensive and time-consuming.

---

> > ### Author Response · Authors · 2021-11-19
> > **Response to Reviewer Ap2b (Part 2)**
> >
> > ***The experiments are OK but not probably not good enough to support the general claims about lifelong learning. E.g., it would be great to show more than 3 tasks.***
> >
> > Thanks for your advice! Please allow us to clarify and address your comment.
> >
> > 1. First, one crucial (but easy to overlook) point in our work is that LFLL not only focuses on learning from different domains, but also from different types of tasks. Considering these two dimensions of LFLL, we have actually investigated **9 tasks** in total.
> > 2. In addition, to evaluate whether our method can perform better than the baselines when learning from a large number of different domains, we consider 5 NLI tasks as classification and combine them with the original 4 classification tasks in our work to form a long sequence of **9 classification** tasks. Then we evaluate LFPT5, Prompt tuning with MAS and Fine tuning with MAS (as MAS performs better than EWC on classification) on this long sequence. The accuracy after learning all tasks is shown below.
> >
> > |  Method   | LFPT5   | Prompt tuning with MAS| Fine tuning with MAS|
> > |  ----  | ----  | ----  | ----  |
> > | Accuracy (%) | $\textbf{43.98}_{\pm 2.68}$ | $9.37_{\pm 3.17}$  | $34.37_{\pm 6.21}$ |
> >
> >
> > From the results, we can observe that our method still performs much better than previous baselines when learning from a large number of tasks. This verifies that our method can perform well in various classification tasks including NLI and demonstrates the effectiveness of our method in handling long sequences of tasks.
> >
> >
> >
> > ***The key part that I did not understand is why prompt tuning isn't already a sufficient life-long learning method. My understanding of prompt tuning is that no model weights are modified, only the prompt is optimized, and optimized prompts are stored for each task. Then given a new task, a new prompt can be trained and stored. This works because the size of prompts is many orders of magnitude smaller than the size of the model. Maybe another way of asking this question is, how is the following different from regular prompt tuning: "while adapting to a new task type, LFPT5 includes and tunes additional prompt embeddings for the new task"?***
> >
> > Please see **Our design details** above
> >
> >
> >
> >
> >
> > ***Another part I do not understand is why pseudo samples need to be generated. Why can't you just use the prior examples?***
> >
> > 1. It may not be feasible to expect that we will always have access to examples of the learned tasks. **This is the setting (and main motivation) of lifelong/continual learning.**
> > 2. The human brain contains lots of knowledge which can be replayed later when needed. We also expect that the model can store and replay the acquired knowledge. In this work, we utilize the generative ability of the languege model to generate pseudo samples which help remember previously learned knowledge. See also [1,2].
> >
> >
> > ***I am missing why this idea of retraining on previous domains each time is a good one. Let's say we're up to 1000 tasks. How much weight does the new task get in this mixture? Won't this take a long time?***
> >
> > Retraining on previous domains (or pseudo samples replay) could help improve the performance of LFLL a lot (comparing LFPT5 with prompt tuning with EWC/MAS), which verifies that the generated pseudo samples contain quite useful information of learned domains. So we adopt the idea of retraining. Although we do retraining, it will not take a long time as we sample only few pseudo examples of previously learned domains.
> >
> > Taking classification as an example, we cannot know in advance how much weight we should put on the new task as it depends on the class number of the new and all learned tasks. However, we can ensure that the sample number per class of the new task (such as 16) is larger than that of the learned tasks (such as 2). So the model can still learn the new task well.
> >
> > ***The introduction could make it more clear why the second two limitations of LLL are indeed limitations. Why NER in particular, instead of other tasks that haven't been explored thoroughly? The negative transfer point needs some more evidence as well, as I believe the decaNLP challenge showed positive transfer, right?***
> >
> > 1. We take NER as an example of sequence labeling task (in addition to classification and generation), as it is one of most popular semantic tagging tasks.
> > 2. Positive transfer and negative transfer are both possible in multitask learning. It depends on the types and domains of the tasks. The evidence of negative transfer is shown in Section 4.4. The comparison between Multitask Prompt Tuning and LFPT5 indicates that the learning of one task could cause negative effect on the learning of another task.
> >
> > ***Is the finetuned BERT-Large a few-shot finetuned or with the full dataset?***
> >
> > BERT-Large is finetuned on few-shot sets. We compare single task few-shot results in Section 4.2
> >
> > ***Other suggestions***
> >
> > We will revise the paper following these suggestions as soon as possible.

---

> > > ### Author Response · Authors · 2021-11-19
> > > **Response to Reviewer Ap2b (Part 3)**
> > >
> > > [1] Sun, Fan-Keng, Cheng-Hao Ho, and Hung-Yi Lee. "LAMOL: LAnguage MOdeling for Lifelong Language Learning." ICLR 2020
> > >
> > > [2] Shin H, Lee J K, Kim J, et al. "Continual learning with deep generative replay." NeurIPS 2017

---

> > > ### Comment · Reviewer_Ap2b · 2021-11-19
> > > **Reluctantly raising my score from 3 to 5. AC: if other reviewers all think it's good, feel free to ignore my review**
> > >
> > > This paper seems like a well-intentioned effort. The authors claim I have grossly misunderstood it, and maybe I have. I hope the area chair will override my score if the other reviewers think it is a strong paper. Unfortunately I do not have time to do an extremely detailed back and forth with the authors during the rest of the response period. So I will just leave the following comments.
> > >
> > > The LAMOL reference made me feel better about why you have to regenerate pseudodata that you once had access to, but I still don't buy why this case matters. Text data is so cheap to store on a hard disk. You might improve your paper by spending 2-3 sentences motivating and explaining this.
> > >
> > > I spent ten minutes trying to understand your explanation of prompt tuning, but I could not understand it. Could you please just clarify why using prompt tuning to learn a new set of embeddings for each task does not satisfy life-long learning? For clarify, I am refering to the prompt tuning of Lester et al. 2021. They can use a single model with the same weights to do any task, only changing the weights of the prompt per task. Hence, the performance of old tasks will not be affected by new tasks.
> > >
> > > I am not convinced by the reasons why the full data setting is not explored.
> > >
> > > For instance, "we do not focus on full data setting as this is not how humans learn" is not a good reason. Lots of things take humans thousands of times to get right. Lots of things humans can do immediately. So whether humans can do it is really irrelevant to whether it is a setting that ought to be explored.
> > >
> > > And for the tasks that you're doing, you have the large labeled datasets, so I don't buy the second excuse later.
> > >
> > > Despite the fact that I'm raising my score from 3 to 5, I am not very convinced by the response. The reason that I am raising my score to 5 is that I may indeed have misunderstood the paper, and I do not have more time to read the paper in more detail (apologies...). But I think the authors allow more people to understand their paper if they explain the above points more clearly.

---

> > ### Comment · Reviewer_Ap2b · 2021-11-19
> > **Maybe I'm wrong**
> >
> > Thanks for the detailed response. I'm surprised that you seriously consider your work "as a vital step towards general language intelligence". "vital" means "absolutely necessary". So if your paper did not come out, then we will not reach general language intelligence? This type of language in my view is what would be used to describe seminal work. I also think people might not agree on the definition of general language intelligence. But in any case, I simply don't see how you can support this claim, or why it's not better to tone down the language a bit instead of trying to support this claim.
> >
> > Maybe I'm wrong and it's not productive to have a back and forth here. If the AC and other reviewers really see LFPT5 as a "vital step towards general language intelligence", I will happy cede my credibility.

---

> > > ### Author Response · Authors · 2021-11-19
> > > **Regarding "Vital" claim**
> > >
> > > Thanks for your reply. We have pointed out that this is the first work towards lifelong few-shot language learning; we formally define the problem and provide a SoTA model and benchmark for future research in this direction. As authors we believe this is a vital step. We will be happy to tone down our claim and replace "vital" with "important".

---

> > > > ### Comment · Reviewer_Ap2b · 2021-11-19
> > > > **Thanks and a nit**
> > > >
> > > > Nit that will not affect my review score: if your work is the first, then it obviously will be SoTA because no one else has done it before right?

---

> > > > > ### Author Response · Authors · 2021-11-19
> > > > > **Clarifying "SoTA"**
> > > > >
> > > > > Thanks again! By "SoTA model", we refer to LFPT5 that extends the recently proposed prompt tuning with language modeling loss and KL divergence loss. We wanted to distinguish from the fact that other methods compared in our work including the existing lifelong learning methods could also be SoTA for the newly defined LFLL problem, if we had not proposed LFPT5.

---

### Official Review · Reviewer_jNL2 · 2021-11-01

**Correctness:** 3
**Technical Novelty And Significance:** 3
**Empirical Novelty And Significance:** 3
**Recommendation:** 6
**Confidence:** 4

**Main Review:**

The paper has several strengths that make it a promising submission:

1. The idea of using prompt-tuning for life-long language learning is a promising direction and the methods proposed in the paper constitute a novel and effective way in bridging the gap between these 2 methodologies. The discussion on few-shot learning also clearly demonstrated the difference of the proposed methods and a major collection of prior work using prompt-tuning.

2. The methods proposed in the paper are effective in utilizing the assumption of language modeling, such that tasks are formed as prompt-label pair sequences, and autoregressive sequence generation is used to anchor the model to previously learnt knowledge.

3. The experimental setup is representative of actual uses of life-long language learning, with 3 different tasks included. Comparisons to fine-tuning, prompt tuning, and their combinations with elastic weight consolidation (EWC) and memory-aware synapses (MAS) are solid choices to demonstrate the effectiveness of the proposed methods. In most of these comparisons, the proposed methods achieved better results.

There are a few weaknesses that I hope the paper can address to improve the scoring:

1. In the paragraph below equation 4, the loss form is an addition of 3 values: 1) the task loss L^{task}; 2) the language modeling loss L^{lm}; 3) label consistency loss L^{KL}. Both 1) and 2) are reasonable, but 3) is new in the paper and it is unclear why it is absolutely necessary. In particular, the paper should include an ablation study on a range of \lambda_{KL}, not just a comparison between models with and without L^{KL}.

2. Some statistics that are useful for understanding the propose methodology are missing in the experimental section: 1) it is probably beneficial to include some numbers on the state-of-the-art results on methods for the original tasks and datasets. They may not be fair statistics in a life-long scenario, but they are helpful for the readers to understand where the methods stand in a larger picture of machine learning methods. 2) For the domain order experiments in section 5, it could be useful to also include accuracy for each of the datasets separately. This will provide the readers with additional knowledge on the relationship between task ordering and individual task performance.

Finally, some suggestions to the paper, but these will not affect the scoring:

1. Remove the quote in page 1.

2. The paper should include a brief introduction to the 2 regularization-based life-long learning methods that it compares with. These are elastic weight consolidation (EWC) and memory-aware synapses (MAS).

3. Try to reduce the use of abbreviations in tables. For example, by typing fine-tuning and prompt-tuning instead of FT and PT, it could make the table much easier to read without the need to refer to the paper text.

**Summary Of The Paper:**

The paper extended the use of prompt tuning to the scenario of life-long language learning. To prevent catastrophic forgetting, the paper proposes to generate pseudo samples as part of the training data when adapting to a new domain, and adding new set of prompt tokens when adapting to a new task. Using the T5 model, the paper demonstrated the effectiveness of the proposed methods using text classification, named entity recognition (NER), and summarization tasks. The method achieved better results for these tasks using the proposed life-long learning mechanisms, compared to fine-tuning, prompt tuning, and their combinations with regularization-based life-long learning methods such as elastic weight consolidation (EWC) and memory-aware synapses (MAS).

**Summary Of The Review:**

Good novelty in extending prompt-tuning to life-long language learning using the generative properties of language models. Good experiments demonstrating the effectiveness of the methods in preventing catastrophic forgetting. Some parts of the method are not sufficiently studied in the experiments.

---

> ### Author Response · Authors · 2021-11-19
> **Response to Reviewer jNL2 (Part 1)**
>
> Thank you for your review! We are encouraged that you found our idea promising, our method novel and effective, our experimental setup representative and our discussion clear. We address your concerns here; we would appreciate your quick response so that we can address further concerns (if any) within the rebuttal time period.
>
> ***In the paragraph below equation 4, the loss form is an addition of 3 values: 1) the task loss $\mathcal L^{task}$; 2) the language modeling loss $\mathcal L^{lm}$; 3) label consistency loss $\mathcal L^{KL}$. Both 1) and 2) are reasonable, but 3) is new in the paper and it is unclear why it is absolutely necessary. In particular, the paper should include an ablation study on a range of $\lambda_{kl}$, not just a comparison between models with and without $\mathcal L^{KL}$.***
>
> Thanks for your advice! The intuition for $\mathcal L^{\text{KL}}$ is that we want the current (student) model to perform more similarly as the previous (teacher) model so that it can retain the learned knowledge. In our current experiments, we simply set $\lambda_{kl}$  to a certain number (e.g. 0.01 for classification and 0.04 for summarization) without tuning such that $\mathcal L^{\text{KL}}$ has roughly the same order of magnitude as the other losses. We have conducted new experiments with different $\lambda_{kl}$ on summarization (since this is faster) and present the results below.
>
> |$\lambda_{kl}$|0|0.02|0.04|0.10|0.20|0.40|
> |  ----  | ----  |  ----  |----  | ----  | ----  | ----  |
> |  A-RG | $16.27_{\pm 0.50}$ | $16.41_{\pm 0.27}$ | $17.05_{\pm 0.92}$ | $\textbf{17.11}_{\pm 0.59}$ | $16.97_{\pm 0.88}$ | $16.26_{\pm 0.32}$ |
>
> As shown in this table, the model achieves the best average ROUGE (A-RG) 17.11 with $\lambda_{kl}$ 0.10 while the average ROUGE is only 16.26 with $\lambda_{kl}$ 0.40. In addition, the performance of the variant without $\mathcal L^{\text{KL}}$ is worse than the performance of all other variants except the variant with $\lambda_{kl}$ 0.40 (too large), which demonstrates the necessity of  $\mathcal L^{\text{KL}}$.
>
>
>
> ***Some statistics that are useful for understanding the propose methodology are missing in the experimental section: 1) it is probably beneficial to include some numbers on the state-of-the-art results on methods for the original tasks and datasets. They may not be fair statistics in a life-long scenario, but they are helpful for the readers to understand where the methods stand in a larger picture of machine learning methods. 2) For the domain order experiments in section 5, it could be useful to also include accuracy for each of the datasets separately. This will provide the readers with additional knowledge on the relationship between task ordering and individual task performance.***
>
> Thanks for your suggestion! We report the new statistics here and will update the paper accordingly.
>
> State-of-the-art results on the original full sets and the single task few-shot results.
>
> **NER and text classification:**
>
> |     | CoNLL03   | OntoNotes | AGNews| Amazon   | DBPedia | Yahoo|
> |  ----  | ----  | ----  | ----  |----  | ----  | ----  |
> |  SOTA| $94.6$ | $92.07$ | $95.55$ |$65.83$ | $99.38$ | $77.62$ |
> |  BERT-Large| $62.67_{\pm 1.34}$  | $\textbf{63.55}_{\pm 1.68}$ | $82.33_{\pm 1.66}$ | $40.47_{\pm 1.39}$  | $97.29_{\pm 0.61}$ | $59.97_{\pm 2.25}$ |
> |  T5 fine-tuning| $53.74_{\pm 1.20}$ | $55.15_{\pm 0.70}$ | $83.17_{\pm 2.60}$ | $\textbf{48.80}_{\pm 2.05}$ | $\textbf{98.19}_{\pm 0.19}$ | $50.07_{\pm 21.84}$ |
> |  T5 prompt tuning| $\textbf{68.40}_{\pm 1.24}$ | $61.23_{\pm 2.14}$ | $\textbf{85.33}_{\pm 1.05}$ |$43.73_{\pm 0.41}$ | $97.36_{\pm 0.52}$ | $\textbf{65.67}_{\pm 2.03}$ |
>
> **Summarization:**
>
> |     |  CNNDM   | WikiHow | Xsum|
> |  ----  | ----  | ----  | ----  |
> |  SOTA| $44.7, 21.4, 41.4$ | $35.91, 13.9, 34.82$ | $47.21, 24.56, 39.25$ |
> |  T5 fine-tuning| $30.62, 10.22, 18.34$ | $\textbf{27.33}, 6.49, 16.32$ | $21.45, 3.99, 14.08$ |
> |  T5 prompt tuning| $\textbf{34.63}, \textbf{14.34}, \textbf{23.70}$ | $26.47, \textbf{8.02}, \textbf{18.78}$ | $\textbf{22.67}, \textbf{5.69}, \textbf{17.36}$ |

---

> > ### Author Response · Authors · 2021-11-19
> > **Response to Reviewer jNL2 (Part 2)**
> >
> > The accuracy for each of the datasets.
> >
> > |  Method   | DB-Amazon-Yahoo-AG   | DB-Amazon-AG-Yahoo | Yahoo-Amazon-AG-DB|
> > |  ----  | ----  | ----  | ----  |
> > |  Prompt tuning | $0.00, 0.00, 7.29, 81.71$ | $0.00, 0.00, 52.57, 64.57$  | $0.71, 0.00, 0.00, 97.86$ |
> > |  Prompt tuning with EWC | $0.00, 0.00, 6.14, 83.14$ | $0.00, 0.00, 52.57, 70.29$  | $0.00, 0.00, 0.00, 98.14$ |
> > |  Prompt tuning with MAS| $0.00, 0.00, 12.57, 83.86$ | $0.00, 0.00, 58.00, 65.71$  | $0.29, 0.00, 0.57, 97.86$ |
> > |  Fine-tuning | $26.71, 6.40, 9.43, 85.71$ | $21.36, 5.40, 57.00, 71.29$  | $4.43, 17.80, 25.86, 98.14$ |
> > |  Fine-tuning with EWC | $38.21, 0.20, 21.00, 86.29$ | $24.07, 4.00, 59.86, 68.14$  | $0.14, 12.80, 5.86, 98.07$ |
> > |  Fine-tuning with MAS| $37.29, 0.60, 15.71, 83.29$ | $20.79, 0.40, 61.14, 67.00$  | $1.00, 11.40, 27.57, 98.07$ |
> > |  LFPT5 | $48.57, 23.20, 32.43, 78.43$ | $54.93, 12.20, 61.86, 67.43$  | $10.57, 9.20, 59.86, 98.00$ |
> > |  Prompt tuning with real data | $85.79, 31.20, 43.14, 82.00$ | $82.57, 37.40, 67.29, 64.43$ | $43.71, 24.20, 58.71, 94.29$|
> > | Multitask prompt tuning  | $95.00, 47.40, 62.29, 75.57$ | $93.57, 47.80, 73.86, 65.57$ | $61.71, 43.00, 74.71, 93.21$ |
> >
> >
> > ***Other suggestions***
> >
> > We will revise the paper following these suggestions as soon as possible.

---

> > > ### Comment · Reviewer_jNL2 · 2021-11-26
> > > **Good additional statistics**
> > >
> > > Thanks to the authors for the additional results!
> > >
> > > The additional experiments on L^{KL} is quite interesting, as it shows that the best choice of \lamda_{KL} should neither be too small (such that it is not useful) or too large (such that the model collapsed and ignored training data). Could you also add this to other tasks? I could imagine this will help those who want to replicate the work and make improvements.
> > >
> > > The additional statistics compared to state-of-the-art should be quite helpful for readers to understand the settings of the model and the regimes in which they are useful.
> > >
> > > On the statistics of different datasets compared to the order of domains, it seems different tuning methods bias the tasks very differently across domains. That may point to different strengths and weaknesses of the models, in particular how much do they forget about the knowledge learnt previously. It feels to me that the paper should give this part more discussion in text since it is closely related to the life-long learning setting mentioned in the paper title.
> > >
> > > In general I feel that the response improved the clarity and significance of the paper, and I'm willing to increase the score to 7 if such an option was offered to reviewers.

---

> > > > ### Author Response · Authors · 2021-11-27
> > > > **Thanks for willing to raise the score to 7**
> > > >
> > > > Thanks for your thoughtful feedback! We are glad that our response and additional experiments were helpful in improving the clarity and significance of the paper. We are also encouraged that you are willing to increase the score to 7.
> > > >
> > > > We are happy to conduct similar experiments with varying $\lambda^{KL}$ on other tasks, and be assured that we will include the results in the final version (in case we cannot finish it within the rebuttal period which will end soon). We will also include further analysis and discussion on different methods in domain order experiments.
> > > >
> > > > Please let us know if there is anything else we can clarify or provide in the meantime.
> > > >
> > > > Best regards

---

### Official Review · Reviewer_PawA · 2021-11-03

**Correctness:** 4
**Technical Novelty And Significance:** 3
**Empirical Novelty And Significance:** 3
**Recommendation:** 6
**Confidence:** 3

**Main Review:**


=================================
Strengths:
  - Combining lifelong and few-shot learning is a new setting.
  - Experiments contain 3 different tasks and each has datasets from different domains.
  - Experiments are well-designed: many baselines are implemented to compare proposed method with traditional lifelong learning methods.
  - Paper is well-written.

=================================
Weaknesses:
  - The actual method is simple combination of existing ideas.
  - The number of tasks and domains is minimal setting. To really become a benchmark to measure the progress of LFLL, more tasks/datasets will be needed.


=================================
Additional Suggestions/Questions:
  - Question Answering and NLI are also very important NLP problems and many datasets exists from different domains. Perhaps the authors can consider adding them to setup a more comprehensive benchmark.
  - Perhaps adding more pre-trained LMs such as GPT-2 and different sizes of T-5. The community will be interested how the model type / scale affect the LFLL capability.
  - How will the pseudo data generation amount affect the learning / forgetting performance? Since the data is generated by T-5, couldn't we generate as much as we want? Maybe adding another ablation on this would be a good idea.




**Summary Of The Paper:**

This paper proposes to leverage prompt tuning on large pre-trained language models to achieve Lifelong Few-shot Language Learning (LFLL). Lifelong learning and few-shot learning have been considered different ML paradigms. The authors argue that with the emergence of modern large-scale pre-trained language models, AI models now have the capability to achieve these 2 at the same time. This work attempts to formally define the LFLL problem and benchmarked a strong pre-trained LM (T5) on 3 tasks (NER, text classification, and summarization) over 9 different datasets (defined as domains in LFLL).

**Summary Of The Review:**

Overall I'm leaning positive about this paper. The proposed LFLL problem, in my opinion, is very important for the community to think about: how to adapt large-scale pre-trained LM to different tasks/domains with few samples? This paper sets up a strong baseline and benchmark for research along this direction. Although prompt tuning, KL regularization, and pseudo data generation are existing ideas, combining them and formulate this LFLL problem/benchmark is non-trivial contribution. I think this work has potential, and if the authors can try to add more tasks/datasets/baselines, it may create more impact to the community.

---

> ### Author Response · Authors · 2021-11-19
> **Response to Reviewer PawA (Part 1)**
>
> Thank you for your comments! We find it encouraging that you think our work is non-trivial contribution and has the potential to create more impact to the community, that LFLL is a very important problem, that experiments are well-designed, and that our paper is well-written. We address your concerns here; we would appreciate your quick response so that we can address further concerns (if any) within the rebuttal time period.
>
> ### Weaknesses:
> ***The actual method is simple combination of existing ideas***
>
> We agree with your opinion that though KL regularization and pseudo data generation are existing ideas, formulating the Lifelong Few-shot Language Learning (LFLL) problem and successfully combining all of them with prompt tuning as a SoTA model for LFLL is non-trival. We believe that with the rise of large-scale pre-trained language models which already contain huge amount of meta knowledge, the community should pay more attention to the challenging yet practical problem: how to efficiently adapt pre-trained language models to new types of tasks and new domains (same type) with only few samples while retaining the knowledge of previously learned tasks. If this problem is successfully solved, we can say that we realize general language intelligence to some extent. Our work takes the first step towards this goal: we formally define this problem as LFLL and provide a SoTA model and benchmark for future research in this direction.
>
> ***The number of tasks and domains is minimal setting. To really become a benchmark to measure the progress of LFLL, more tasks/datasets will be needed. Question Answering and NLI are also very important NLP problems and many datasets exists from different domains. The authors can consider adding them to setup a more comprehensive benchmark.***
>
> Thanks for your suggestion. We agree that setting up a comprehensive benchmark for LFLL is a long-term work that may require a community-wide involvement. We will create a leaderboard for that and keep improving the benchmark.
>
> We have added NLI to our benchmark following your advice. Our experiments on QA are still running; we will add the results as soon as we get the results.
>
> We regard 5 Natural Language Inference (NLI) tasks as classification and combine them with the original 4 classification tasks in our work to form a long sequence of 9 classification tasks. Then we evaluate LFPT5, Prompt tuning with MAS and Fine tuning with MAS (as MAS performs better than EWC on classification) on this long sequence. We also want to use this experiment to verify whether our method can outperform previous baselines when processing a large number of tasks. The accuracy after learning all tasks is shown below.
>
> |Method|LFPT5| Prompt tuning with MAS| Fine tuning with MAS|
> |---- |----|----|----|
> |Accuracy (%)|$\textbf{43.98}_{\pm 2.68}$|$9.37_{\pm 3.17}$|$34.37_{\pm 6.21}$|
>
> From the results, we can observe that our method performs much better than previous baselines when learning from a large number of tasks. This verifies that our method can perform well in various classification tasks including NLI and demonstrates the effectiveness of our method in handling long sequence of tasks.
>
> ***Perhaps adding more LMs such as GPT-2 and different sizes of T-5. The community will be interested how the model type/scale affect the LFLL capability.***
>
> Again thanks for your suggestion! As a proof of concept, we have added an experiment with T5-base on summarization (as it is faster to run) and will continually add more diverse model types/scales. We compare the results (average ROUGE) of LFPT5, Prompt tuning with EWC and Fine tuning with EWC in the following table. We can see that LFPT5 performs much better than Prompt tuning with EWC. However, it is slightly worse than Fine tuning with EWC. This is consistent with the finding in [1] that prompt tuning performs better when applied to larger pretrained language models.
>
> |Method|LFPT5|Prompt tuning with EWC|Fine tuning with EWC|
> |----|----|----|----|
> |A-RG|$14.93_{\pm 0.82}$|$13.05_{\pm 1.23}$|$\textbf{15.15}_{\pm 1.38}$|
>
> ***How will the pseudo data generation amount affect the learning / forgetting performance? Since the data is generated by T5,  couldn't we generate as much as we want? Maybe adding another ablation on this would be a good idea.***
>
> Genarating pseudo samples is feasible and cheap. We can use any number of pseudo samples. We have conducted an experiment to analyze the influence of different numbers of pseudo samples, as shown below.
>
> |Number|2|4|8|16|32|
> |----|----|----|----|----|----|
> |A-RG|$16.28_{\pm 0.95}$|$\textbf{17.05}_{\pm 0.92}$|$16.80_{\pm 0.52}$|$16.74_{\pm 0.45}$ | $16.79_{\pm 0.64}$|
>
> We can observe that increasing the number of pseudo samples will not always improve the performance. The model achieves the best average ROUGE 17.05 with 4 pseudo samples.
>
> [1] Lester, Brian, Rami Al-Rfou, and Noah Constant. "The power of scale for parameter-efficient prompt tuning." EMNLP 2021

---

> > ### Author Response · Authors · 2021-11-19
> > **Response to Reviewer PawA (Part 2)**
> >
> > [1] Lester, Brian, Rami Al-Rfou, and Noah Constant. "The power of scale for parameter-efficient prompt tuning." EMNLP 2021

---

> > ### Author Response · Authors · 2021-11-29
> > **Results on Question Answering.**
> >
> > We have added the following key results on Question Answering (QA) or Machine Reaching Comprehension (MRC) to our benchmark following your advice. Specifically, we evaluate LFPT5, Prompt tuning with EWC and Fine-tuning with EWC on a sequence of 3 MRC tasks: SuperGLUE-ReCoRD [1], SQuAD v1.1 [2] and HotpotQA [3]. In all cases, we have 32 training examples (32-shot). The average results (F1 score) of three different runs (different task order and few-shot data) are shown below.
> >
> > |  Method  | LFPT5  | Prompt tuning with EWC| Fine-tuning with EWC|
> > |  ----  | ----  | ----  | ----  |
> > | F1| $\textbf{30.63}_{\pm 7.08}$ | $28.50_{\pm 11.13}$  | $15.84_{\pm 11.63}$ |
> >
> > From the results, we can observe that LFPT5 also outperforms previous baselines when learning on MRC tasks.
> >
> >
> >
> > [1] Wang A, Pruksachatkun Y, Nangia N, et al. "SuperGLUE: A Stickier Benchmark for General-Purpose Language Understanding Systems." NeurIPS 2019
> >
> > [2] Rajpurkar P, Zhang J, Lopyrev K, et al. "SQuAD: 100,000+ Questions for Machine Comprehension of Text." EMNLP 2016
> >
> > [3] Yang Z, Qi P, Zhang S, et al. "HotpotQA: A Dataset for Diverse, Explainable Multi-hop Question Answering." EMNLP 2018

---

> > ### Comment · Reviewer_PawA · 2021-12-06
> > **Thanks for the response**
> >
> > Thank you for the detailed response. I really appreciate the effort of additional experiments. Since the authors' response and revision addresses my concern, I recommend accept for this paper and update my overall score to 7 if allowed (the system does not have this option).

---

> > > ### Author Response · Authors · 2021-12-07
> > > **Thanks for willing to raise the score to 7**
> > >
> > > Thanks for your feedback and we are happy that you are convinced with our response and the revised version!

---

### Official Review · Reviewer_9Wdo · 2021-11-04

**Correctness:** 3
**Technical Novelty And Significance:** 2
**Empirical Novelty And Significance:** 3
**Recommendation:** 5
**Confidence:** 5

**Main Review:**

Strengths:

- The paper attempts to solve an important problem of LFLL in the context of the pre-trained Transformers. With the rise of larger models, it is important to consider the scalability of these models for lifelong learning. Given that this paper builds upon the recent developments in parameter-efficient transfer learning, the number of parameters in the proposed prompt tuning-based method would not grow dramatically with the number of tasks.
- Empirical results showcase that the proposed approach is superior to existing methods like EWC and MAS when evaluated on the sequence of 2-4 domains/tasks.


Weaknesses:

- For the LFLL problem, this paper proposes to use an existing prompt tuning-based transfer learning method where the underlying model is frozen and a few additional task/domain-specific parameters are learned. Furthermore, it repurposes existing lifelong learning methods like deep generative replay [1], distillation-based methods [2], dynamically expandable network [3] as a unified framework when applied to prompt tuning. Thus, the contributions of this paper are marginally novel/ significant.
- Another major limitation of this paper is that it completely ignores comparison with other parameter-efficient transfer learning methods like adapters [4]. Concretely, AdapterFusion [3] work proposes to learn a task-specific composition of adapters from previous tasks. How does the prompt tuning-based method compare with adapter fusion when applied to the LFLL problem? Is there a consensus that prompt tuning-based methods are better than adapters for few-shot learning?
- By definition, the lifelong learning paradigm deals with a large number of tasks in sequence. However, the paper evaluates the proposed approach on the sequence of 2-4 tasks. So it is unclear whether the gains persist as we increase the number of tasks to 10-15 (towards a more realistic evaluation setup). As we increase the number of tasks, does it enable positive forward/ backward transfer or increase the chance of negative transfer?
- Overall, the paper is written in good style. However, there are key experimental details that are missing. This paper defines prompts as a series of tunable tokens but then it is unclear how many tokens are defined per task prompt. This is important because as we scale the number of tasks, the input sequence length scales linearly, and with that self-attention scales quadratically. This issue limits the practical applicability of the prompt tuning-based method for lifelong learning.
- What does it mean to use validation sets for hyper-parameter selection? Ideally, during lifelong learning, we do not know the sequence of tasks apriori. It is unfair to freeze a particular task order, then search for optimal hyper-parameters, and then report results on multiple runs with the same task order. At least, report the results on unseen task orders? Or search on a subset of disjoint tasks [5]. The paper should consider updating their results with a fair hyper-parameter selection strategy.

Additional clarification question(s):
- The paper trains a single prompt for both task solving and data generation. It is unclear whether this is an optimal strategy when it comes to generating pseudo-samples? Did the paper try learning a separate prompt for task solving and data generation? Does it help in addressing the quality of generated pseudo-samples? Furthermore, how effective is the current pseudo-sample generation strategy? How many samples are discarded to get valid samples?
- There are a few hyper-parameters that are set without any justification: 16-shot learning? How does the method perform with varying the number of training examples? How does the performance change with varying the number of pseudo-samples? Does it hurt if we have more pseudo-samples (generating them is feasible and cheaper)?
- In terms of the multi-task experiments (MT-PT), do we have separate prompts or a single prompt being trained with all data? If it is a single prompt then is it fair to compare it with LFPT5 with more capacity? If there are multiple prompts then how does one ensure that there are dedicated prompts for each task?

[1] Sun, Fan-Keng, Cheng-Hao Ho, and Hung-Yi Lee. "LAMOL: LAnguage MOdeling for Lifelong Language Learning." International Conference on Learning Representations. 2020.

[2] Li, Zhizhong, and Derek Hoiem. "Learning without forgetting." IEEE transactions on pattern analysis and machine intelligence 40.12 (2017): 2935-2947.

[3] Pfeiffer, Jonas, et al. "AdapterFusion: Non-Destructive Task Composition for Transfer Learning." Proceedings of the 16th Conference of the European Chapter of the Association for Computational Linguistics: Main Volume. 2021.

[4] Houlsby, Neil, et al. "Parameter-efficient transfer learning for NLP." International Conference on Machine Learning. PMLR, 2019.

[5] Chaudhry, Arslan, et al. "Efficient Lifelong Learning with A-GEM." International Conference on Learning Representations. 2018.


**Summary Of The Paper:**

The paper defines the problem of lifelong few-shot language learning (LFLL) where the goal is to continuously learn a model with new few-shot tasks without forgetting the previous ones. Towards this problem, the paper introduces a prompt tuning-based framework that augments the pre-trained language model (T5) with continuous prompts (trainable prompt embeddings). Prompts are simultaneously optimized for task solving and data generation. During continual learning of the new domains, pseudo-examples of the previously seen domains are generated for episodic rehearsal and further KL-based consistency regularization is implemented to prevent drifting of the model. Lastly, for new tasks, separate prompts are appended with the existing prompts to enable the transfer of knowledge from previous domains/ tasks. The main contribution of this paper is to showcase the effectiveness of the recently proposed prompt tuning-based method for lifelong language learning. Empirically, this paper reports superior results over EWC and MAS methods when evaluated on text classification, NER, and summarization tasks.

**Summary Of The Review:**

Overall, I rate this paper marginally below the acceptance threshold. It is interesting to see the efficacy of prompt tuning for lifelong language learning. However, with the current experiments, it is unclear how they compare with adapters? Also given this paper unifies existing approaches and adapts them for prompt tuning setup, it is marginally significant/ novel. There are several open questions (see weakness section + additional clarification questions) that need to be answered before understanding the overall benefits of the proposed framework. I am looking forward to the author's response in the rebuttal period and will consider updating my scores accordingly.

---

> ### Author Response · Authors · 2021-11-19
> **Response to Reviewer 9Wdo (Part 1)**
>
> Thank you for your comments! We are glad that you found our approach superior to existing methods and parameter-efficient, our paper well-written, and recognized the importance of LFLL. We address your concerns here; we would appreciate your quick response so that we can address further concerns (if any) within the rebuttal time period.
>
> ### Weakness
>
> ***For the LFLL problem, this paper proposes to use an existing prompt tuning-based transfer learning method where the underlying model is frozen and a few additional task/domain-specific parameters are learned. Furthermore, it repurposes existing lifelong learning methods like deep generative replay, distillation-based methods, dynamically expandable network as a unified framework when applied to prompt tuning. Thus, the contributions of this paper are marginally novel/ significant.***
>
> We would like to argue that though deep generative replay, distillation-based methods and dynamically expandable networks are existing ideas, formulating the Lifelong Few-shot Language Learning (LFLL) problem and successfully combining all of them with prompt tuning as a strong baseline for LFLL is non-trival. We believe that with the rise of large-scale pre-trained language models which already contain huge amount of meta knowledge, the community should pay more attention to the challenging yet practical promlem: how to efficiently adapt pre-trained language models to new types of tasks and new domains (same type) with only few samples while retaining the knowledge of previously learned tasks. If this problem is successfully solved, we can say that we realize general language intelligence to some extent. Our work takes the first step towards this goal: we formally define this problem as LFLL and provide a SoTA model and benchmark for future research in this direction.
>
>
> ***Another major limitation of this paper is that it completely ignores comparison with other parameter-efficient transfer learning methods like adapters. Concretely, AdapterFusion work proposes to learn a task-specific composition of adapters from previous tasks. How does the prompt tuning-based method compare with adapter fusion when applied to the LFLL problem? Is there a consensus that prompt tuning-based methods are better than adapters for few-shot learning?***
>
> Thanks for your suggestion. We were not aware of this recent work from EACL-2021 before. We have performed new experiments to compare AdapterFusion [1] with LFPT5 when the model learns from different types of tasks. The results of three different runs (different task order and few-shot data) are shown below.
>
> |Method| Summ-Class-NER| Class-NER-Summ| NER-Summ-Class|
> |----|----|----|----|
> | AdapterFusion | $22.26, 83.25, 55.62$ | $81.25, 63.19, 22.37$  | $62.99, 21.20, 82.50$ |
> | LFPT5 w.o. FKT  | $\textbf{25.48}, 84.75, \textbf{63.28}$ | $\textbf{83.25}, \textbf{67.66}, 23.68$ | $\textbf{66.65}, \textbf{22.97}, \textbf{84.50}$|
> | LFPT5 with FKT  | $\textbf{25.48}, \textbf{86.00}, 62.44$ | $\textbf{83.25}, 65.01, \textbf{24.92}$ | $\textbf{66.65}, 22.80, 84.25$ |
>
> From the results, we can see that although there is no consensus or theoretical proof that prompt tuning-based methods are better than adapters for few-shot learning, LFPT5 performs better than AdapterFusion in LFLL settings which demonstrate the superiority of our method. Moreover, our method is much more parameter-efficient than AdapterFusion. In the course of learning these three tasks, AdapterFusion introduces about 21.72% of the parameters of the pretrained model, while our method only adds about 0.12%.
>
>
> ***By definition, the lifelong learning paradigm deals with a large number of tasks in sequence. However, the paper evaluates the proposed approach on the sequence of 2-4 tasks. So it is unclear whether the gains persist as we increase the number of tasks to 10-15 (towards a more realistic evaluation setup). As we increase the number of tasks, does it enable positive forward/ backward transfer or increase the chance of negative transfer?***
>
> Thanks for your advice! Please allow us to clarify and address your comment.
>
> 1. First, one crucial (but easy to overlook) point in our work is that LFLL not only focuses on learning from different domains, but also from different types of tasks. Considering these two dimensions of LFLL, we have actually investigated **9 tasks** in total.
> 2. In addition, to evaluate whether our method can perform better than the baselines when learning from a large number of different domains, we consider 5 NLI tasks as classification and combine them with the original 4 classification tasks in our work to form a long sequence of **9 classification** tasks. Then we evaluate LFPT5, Prompt tuning with MAS and Fine tuning with MAS (as MAS performs better than EWC on classification) on this long sequence. The accuracy after learning all tasks is shown below.

---

> > ### Author Response · Authors · 2021-11-19
> > **Response to Reviewer 9Wdo (Part 2)**
> >
> > |Method|LFPT5|Prompt tuning with MAS| Fine tuning with MAS|
> > |----|----|----|----|
> > |Accuracy (%)| $\textbf{43.98}_{\pm 2.68}$ | $9.37_{\pm 3.17}$  | $34.37_{\pm 6.21}$ |
> >
> > From the results, we can observe that our method still performs much better than previous baselines when learning from a large number of tasks.
> >
> > It's hard to determine whether learning a new task will enable positive forward/backward transfer or increase the chance of negative transfer as the order of tasks is unknown. We cannot know the relationship between     the new task and the previously learned tasks. But we empirically analyze that as we increase the number of tasks, it will be more difficult to share knowledge among a larger number of tasks. The learning of new         tasks is more likely to erase the knowledge of previous tasks that are incompatible with new tasks.
> >
> >
> > ***Overall,the paper is written in good style. However, there are key experimental details that are missing. This paper defines prompts as a series of tunable tokens but then it is unclear how many tokens are defined per task prompt. This is important because as we scale the number of tasks, the input sequence length scales linearly, and with that self-attention scales quadratically. This issue limits the practical applicability of the prompt tuning-based method for lifelong learning.***
> >
> > Please allow us to clarify:
> >
> > 1. We define **300 tunable tokens** per task prompt. So we will add about 0.04% of the parameters of the pretrained model when learning from a new task type (e.g., Classification, Generation).
> > 2. We believe there could be a misunderstanding on the usage of tunable prompts. (A) When learning new domains of the same task (e.g., NER of CoNLL and OntoNotes), we share the same (300) prompt tokens, i.e. the same token embeddings get updated as we go from one to a new domain. (B) When learning a new task type (e.g. NER, Generation), we use a different set of (300) prompt tokens. The new prompt tokens are initialized from the vocabulary of T5 or the prompt tokens of the previously learned task as described in Section 4.4.
> > For each task, we will concat the prompt tokens for its type and the input data. So the input sequence length of this task is the sum of prompt length (300) and input data length. In other words, the prompt length of each task is fixed and it does not increase in either dimension of LFLL.
> >
> > ***What does it mean to use validation sets for hyper-parameter selection? Ideally, during lifelong learning, we do not know the sequence of tasks apriori. It is unfair to freeze a particular task order, then search for optimal hyper-parameters, and then report results on multiple runs with the same task order. At least, report the results on unseen task orders? Or search on a subset of disjoint tasks. The paper should consider updating their results with a fair hyper-parameter selection strategy.***
> >
> > Please allow us to clarify:
> >
> > The hyper-parameter selection process is done only when comparing the single task few-shot results in Section 4.2. We do not conduct hyper-parameter search during the LFLL process. For example, we select the optimal hyper-parameters (learning rate 0.5 and LM loss weight 0.25) for AGNews and then use these hyper-parameters during the LFLL process without tuning. We do not assume any prior knowledge of the task order and report the average results of three different runs with different task order. So, we think the comparison is fair.
> >
> > For the weight $\lambda_{kl}$ of the KL divergence loss $\mathcal L^{\text{KL}}$, we simply set its value to a certain number (such as 0.01 for classification and 0.04 for summarization) without tuning such that the KL loss has roughly the same order of magnitude as the other losses. To further investigate, we have conducted new experiments with different $\lambda_{kl}$ on summarization. The results are shown below.
> >
> > |$\lambda_{kl}$|0|0.02|0.04|0.10|0.20|0.40|
> > |  ----  | ----  |  ----  |----  | ----  | ----  | ----  |
> > |  A-RG | $16.27_{\pm 0.50}$ | $16.41_{\pm 0.27}$ | $17.05_{\pm 0.92}$ | $\textbf{17.11}_{\pm 0.59}$ | $16.97_{\pm 0.88}$ | $16.26_{\pm 0.32}$ |
> >
> > As shown in this table, the model achieves the best average ROUGE (A-RG) of 17.11 for $\lambda_{kl}$ 0.10 while the average ROUGE is 16.26 for $\lambda_{kl}$ 0.40. The performance of the variant without $\mathcal L^{\text{KL}}$ (i.e., $\lambda_{kl} = 0$)  is worse than the performance of all other variants except the variant with $\lambda_{kl}$ 0.40 (too large). This demonstrates the effectiveness of $\mathcal L^{\text{KL}}$.

---

> > > ### Author Response · Authors · 2021-11-19
> > > **Response to Reviewer 9Wdo (Part 3)**
> > >
> > > ***The paper trains a single prompt for both task solving and data generation. It is unclear whether this is an optimal strategy when it comes to generating pseudo-samples?  Did the paper try learning a separate prompt for task solving and data generation?  Does it help in addressing the quality of generated pseudo-samples? Furthermore, how effective is the current pseudo-sample generation strategy? How many samples are discarded to get valid samples?***
> > >
> > > 1. Thanks for your suggestion. We have conducted new experiments on summarization (since it is faster) to explore whether learning separate prompts for task solving and data generation would be better. We compare the performance of a single prompt (300 tokens) and two separate prompts (150 tokens for task solving and 150 tokens for data generation) in LFLL settings (average of three different runs with different task order). The results are shown below.
> > > |  Method   | single prompt   | separate prompts|
> > > |  ----  | ----  | ----  |
> > > | ROUGE score | $24.91_{\pm 1.51}, 7.92_{\pm 0.55}, 18.31_{\pm 0.72}$ | $24.89_{\pm 0.56}, 7.82_{\pm 0.22}, 18.27_{\pm 0.33}$  |
> > >
> > > We can observe that although learning separate prompts for task solving and data generation reduces variance, the performance drops slightly. From this experiment, it appears that the quality of pseudo samples generated by these two methods is somewhat similar. We can include more experiments and report in our paper, if needed.
> > >
> > > 2. We generate 1000 pseudo samples using our current pseudo sample generation strategy. There are on average around 278 valid samples. We believe there remains room for improvement in the pseudo sample generation process.
> > >
> > > ***There are a few hyper-parameters that are set without any justification: 16-shot learning? How does the method perform with varying the number of training examples?  How does the performance change with varying the number of pseudo-samples? Does it hurt if we have more pseudo-samples (generating them is feasible and cheaper)?***
> > >
> > > 1. Our choice of 16-shot learning for NER and summarization is based on the settings of [2]. This amounts to at least 64 samples in the training/validation set (the minimum class number is 4). So, we sample 64 examples for the training/validation set per domain for summarization.
> > >
> > > We have also conducted new experiments to compare the performance of LFPT5, Prompt tuning with EWC and Fine tuning with EWC (as EWC performs better than MAS on summarization) with different numbers (16, 32) of few-shot data on summarization. The results (average ROUGE) are shown below.
> > >
> > > **16-shot:**
> > >
> > > |  Method  | LFPT5   | Prompt tuning with EWC| Fine tuning with EWC|
> > > |  ----  | ----  | ----  | ----  |
> > > | A-RG | $\textbf{15.11}_{\pm 0.44}$ | $14.16_{\pm 0.42}$  | $13.75_{\pm 2.35}$ |
> > >
> > > **32-shot:**
> > >
> > > |  Method  | LFPT5   | Prompt tuning with EWC| Fine tuning with EWC|
> > > |  ----  | ----  | ----  | ----  |
> > > | A-RG | $\textbf{15.58}_{\pm 0.27}$ | $14.30_{\pm 0.48}$  | $14.78_{\pm 1.49}$ |
> > >
> > > From the results, we can see that our method consistently outperforms previous baselines with different numbers of few-shot samples.
> > >
> > > 2. Generating pseudo samples is feasible and cheaper. We can use any number of pseudo samples. We have conducted new experiments on summarization to analyze the influence of different numbers of pseudo samples.
> > > |  Number  | 2 |4 |8 | 16 |32 |
> > > |  ----  | ----  | ----  | ----  |----  | ----  |
> > > | A-RG | $16.28_{\pm 0.95}$ | $\textbf{17.05}_{\pm 0.92}$  | $16.80_{\pm 0.52}$ | $16.74_{\pm 0.45}$ | $16.79_{\pm 0.64}$|
> > >
> > > We can observe that increasing the number of pseudo samples will not always improve the performance. The model achieves the best average ROUGE 17.05 with 4 pseudo samples.

---

> > > > ### Author Response · Authors · 2021-11-19
> > > > **Response to Reviewer 9Wdo (Part 4)**
> > > >
> > > > ***In terms of the multi-task experiments (MT-PT), do we have separate prompts or a single prompt being trained with all data? If it is a single prompt then is it fair to compare it with LFPT5 with more capacity? If there are multiple prompts then how does one ensure that there are dedicated prompts for each task?***
> > > >
> > > > We use a single prompt for multitask prompt tuning (MT-PT) and report the results in Table 5. We agree that the comparison might be unfair due to different model capacity. To better support our claim, we have conducted new multitask prompt tuning experiments using the same number of tunable tokens as LFPT5. The tunable tokens are shared among all three tasks. From the results shown below, we can see that LFPT5 performs better than both multitask prompt tuning methods.
> > > >
> > > > |  Method   | Summ-Class-NER   | Class-NER-Summ| NER-Summ-Class|
> > > > |  ----  | ----  | ----  | ----  |
> > > > |  Multitask Prompt Tuning (single prompt) | $24.16, 85.50, 50.80$ | $82.75, 65.31, 23.36$  | $62.83, 11.51, 83.25$ |
> > > > |  Multitask Prompt Tuning (multiple prompts) | $21.01, 82.25, 58.59$ | $83.00, 64.83, 22.73$  | $63.23, 21.41, 84.00$ |
> > > > | LFPT5 w.o. FKT  | $\textbf{25.48}, 84.75, \textbf{63.28}$ | $\textbf{83.25}, \textbf{67.66}, 23.68$ | $\textbf{66.65}, \textbf{22.97}, \textbf{84.50}$|
> > > > | LFPT5 with FKT  | $\textbf{25.48}, \textbf{86.00}, 62.44$ | $\textbf{83.25}, 65.01, \textbf{24.92}$ | $\textbf{66.65}, 22.80, 84.25$ |
> > > >
> > > > For MT-PT, we argue that we should not use separate prompts for each task. In prompt tuning, the parameters of the pretrained language model are freezed. If each task optimizes its own prompt embeddings, it becomes single task prompt tuning (i.e., minimum sharing).
> > > >
> > > > Note that our experiment with MT-PT (Table 5) uses shared prompt embeddings. In this experiment, all three tasks optimize the same embedding space. We do not pick a single best model for all tasks. Instead, we choose one model per task based on its performance on the validation set of this task during learning. In this way, there is a dedicated prompt for each task.
> > > >
> > > >
> > > > [1] Pfeiffer, Jonas, et al. "AdapterFusion: Non-Destructive Task Composition for Transfer Learning." Proceedings of the 16th Conference of the European Chapter of the Association for Computational Linguistics: Main Volume. 2021.
> > > >
> > > > [2] Gao, Tianyu, Adam Fisch, and Danqi Chen. "Making pre-trained language models better few-shot learners." ACL 2021

---

> ### Comment · Reviewer_9Wdo · 2021-12-01
> **I am going to hold to my original rating (5: marginally below the acceptance threshold)**
>
> Thanks to the authors for their detailed response and additional results with AdapterFusion! Please find my final comments below:
>
> 1. In response to the missing experimental/ implementation details, the authors clarify that for new task types (NER, Generation), different sets of prompts are used (in the spirit of existing dynamically expandable networks for lifelong learning). Moreover, for each task, prompt tokens for its type are concatenated with the input and previous task prompts are not considered while learning new tasks. The lifelong learning paradigm focuses on sequential learning of new tasks by enabling forward transfer from previously learned tasks and alleviating forgetting of previous tasks. Given the proposed approach does not consider the previous task prompts, it limits the proposed approach to transfer only from a pre-trained model, which is not different from prompt tuning (similar concerns raised by the reviewer Ap2b).
> 2. In response to the fair comparison between Multitask PT and LFPT5, the authors conduct additional experiments by controlling for the tunable parameters. It is unclear why do authors have different results (significantly different) across multiple sequences..why does task order even matter for multitask learning?
> 3. The paper compares their proposed approach with EWC/ MAS and they report superior performance from their approach. Furthermore, the authors mention that they did not conduct any hyper-parameter search for their method. How and what hyper-parameters do they decide for baseline methods? There is no mention throughout the paper. Surprisingly, the additional results (Table 6) show that performance on previous tasks for EWC/ MAS PT methods is almost 0. Given my experience (with both CV and NLP benchmarks), with very few tasks (like 3-4 considered in this paper), EWC suffers from learning capacity for future tasks but never forgets previous tasks. The reported results are counterintuitive without any discussion. The paper should specify more details around considered baselines in their next iteration. Without that information, it is unclear whether the baselines are properly tuned for the proposed setup. Even AdapterFusion details are missing (e.g., bottleneck size, etc.).
> Moreover, this will also help to clarify their observation (response to the reviewer PawA) -- “We can see that LFPT5 performs much better than Prompt tuning with EWC. However, it is slightly worse than Fine-tuning with EWC. This is consistent with the finding in [1] that prompt tuning performs better when applied to larger pre-trained language models.” The authors might be incorrectly inferring that scale to be the answer to why their method performs better than Fine-tuning with EWC. It might be just sub-optimal hyper-parameter tuning of baselines for T5-Large models.
>
> Based on my above-raised concerns, I am going to hold to my original rating.

---

> > ### Author Response · Authors · 2021-12-02
> > **Response to new concerns (Part 1)**
> >
> > Dear Reviewer,
> >
> > Thanks again for your feedback! Please allow us to address your concern. We hope that you will now be convinced.
> >
> > **1. In response to the missing experimental/ implementation details, the authors clarify that for new task types (NER, Generation), different sets of prompts are used (in the spirit of existing dynamically expandable networks for lifelong learning). Moreover, for each task, prompt tokens for its type are concatenated with the input and previous task prompts are not considered while learning new tasks. The lifelong learning paradigm focuses on sequential learning of new tasks by enabling forward transfer from previously learned tasks and alleviating forgetting of previous tasks. Given the proposed approach does not consider the previous task prompts, it limits the proposed approach to transfer only from a pre-trained model, which is not different from prompt tuning (similar concerns raised by the reviewer Ap2b)**
> >
> > This is actually not the case; some key results might have been oversighted. As mentioned in Sec. 3.2.1 (Page 5) "The embedding of the new prompt tokens can be initialized with the learned embeddings to avoid forgetting and to have a better task prior" and further validated with experiments in Sec. 4.4; see the description and results of **LFPT5 with FKT** and **LFPT5 w.o. FKT** in Page 8 and Table 5 (FKT stands for forward knowledge transfer).
> >
> > Initialization with pretrained models/embeddings has been one of the most successful transfer (and semi-supervised) learning methods. In the context of prompt tuning, several papers have already shown that a good initialization can improve the performance by a good margin [1,2].
> >
> > **2. In response to the fair comparison between Multitask PT and LFPT5, the authors conduct additional experiments by controlling for the tunable parameters. It is unclear why do authors have different results (significantly different) across multiple sequences..why does task order even matter for multitask learning?**
> >
> >
> > We use **different few-shot samples** for different runs. So the different results of multitask learning across multiple sequences are due to different few-shot samples, not due to the task order.
> >
> > **3. The paper compares their proposed approach with EWC/ MAS and they report superior performance from their approach. Furthermore, the authors mention that they did not conduct any hyper-parameter search for their method. How and what hyper-parameters do they decide for baseline methods? There is no mention throughout the paper. Surprisingly, the additional results (Table 6) show that performance on previous tasks for EWC/ MAS PT methods is almost 0. Given my experience (with both CV and NLP benchmarks), with very few tasks (like 3-4 considered in this paper), EWC suffers from learning capacity for future tasks but never forgets previous tasks. The reported results are counterintuitive without any discussion. The paper should specify more details around considered baselines in their next iteration. Without that information, it is unclear whether the baselines are properly tuned for the proposed setup. Even AdapterFusion details are missing (e.g., bottleneck size, etc.). Moreover, this will also help to clarify their observation (response to the reviewer PawA) -- “We can see that LFPT5 performs much better than Prompt tuning with EWC. However, it is slightly worse than Fine-tuning with EWC. This is consistent with the finding in [1] that prompt tuning performs better when applied to larger pre-trained language models.” The authors might be incorrectly inferring that scale to be the answer to why their method performs better than Fine-tuning with EWC. It might be just sub-optimal hyper-parameter tuning of baselines for T5-Large models.**
> >
> > 1. For the base methods, we did conduct hyper-parameter search to get the best average results. Taking fine-tuning with mas on summarization as an example, we run the experiments with different $\lambda_{mas}$ (the weight of $\mathcal L^{\text{mas}}$) and pick the one with the best average results. The results with different $\lambda_{mas}$ we have obtained are shown below.
> >
> >     |  $\lambda_{mas}$  | 0.1  | 0.2 | 0.5| 1.0| 1.5 |2.5 |
> >     |  ----  | ----  | ----  | ----  |----  | ----  | ----  |
> >     | A-RG | $15.64_{\pm 1.86}$ | $15.05_{\pm 3.39}$  | $\textbf{15.76}_{\pm 1.71}$ | $15.66_{\pm 0.65}$ | $15.69_{\pm 1.94}$  | $14.16_{\pm 4.22}$ |
> >
> >     So we choose $\lambda_{mas}$ as 0.5 to get the best average results.
> >
> >     We performed similar hyper-parameter search for EWC. We will put these additional details in the Appendix and improve the discussion of the results.

---

> > > ### Author Response · Authors · 2021-12-02
> > > **Response to new concerns (Part 2)**
> > >
> > > 2. **The additional results (Table 6) show that performance on previous tasks for EWC/ MAS PT methods is almost 0.**
> > >
> > >     We can observe from the table that this is true just for prompt tuning, not for fine-tuning. There could be two reasons:
> > >     1. There are only **a few tunable parameters** in prompt tuning, which is difficult for retaining and accumulating knowledge. So the learning of new knowledge from different domains is more likely to cause the forgetting of previously learned knowledge (explained in "Text Classification" part in page 7). Our method utilizes pseudo samples to alleviate this problem.
> > >     2. We focus on lifelong **few-shot** language learning. The model training is already sub-optimal even without lifelong learning. So the performance is relatively low.
> > >
> > > 3. **AdapterFusion details**: For AdapterFusion, we adopt the implementation from AdapterHub [3] and use the default adapter settings for T5. The default bottleneck reduction factor is 16, i.e., **the bottleneck size is 64**. We adopt a learning rate of 1e-4 with AdamW and a linear learning rate decay following the original AdapterFusion paper [4]. All other hyper-parameter settings (such as batch size and evaluation interval) are the same as LFPT5.
> > >
> > >     We will be happy to add more experiments on different bottleneck sizes, if needed.
> > >
> > > 4. **The authors might be incorrectly inferring that scale to be the answer to why their method performs better than Fine-tuning with EWC. It might be just sub-optimal hyper-parameter tuning of baselines for T5-Large models.**
> > >
> > >     We would like to point out that the result of fine-tuning with EWC reported in Table 4 is already the optimal result after hyper-parameter search. The results with different $\lambda_{ewc}$ (the weight of $\mathcal L^{\text{ewc}}$) are shown below.
> > >
> > >     | $\lambda_{ewc}$  | 0.2  | 0.5 | 1.0| 5.0|
> > >     |  ----  | ----  | ----  | ----  |----  |
> > >     | A-RG | $14.84_{\pm 1.38}$ | $\textbf{15.91}_{\pm 1.46}$  | $15.36_{\pm 1.50}$  | $15.29_{\pm 1.83}$ |
> > >
> > >     We report the result with $\lambda_{ewc}$ 0.5. And our method outperforms this optimal result by a large margin.
> > >
> > > Thanks again for your feedback. We hope that this addresses your concerns. Please be assured that we will include them in the final version.
> > >
> > > [1] Gu Y, Han X, Liu Z, et al. "Ppt: Pre-trained prompt tuning for few-shot learning." arXiv preprint arXiv:2109.04332, 2021.
> > >
> > > [2] Vu T, Lester B, Constant N, et al. "Spot: Better frozen model adaptation through soft prompt transfer." arXiv preprint arXiv:2110.07904, 2021.
> > >
> > > [3] AdapterHub. https://adapterhub.ml/
> > >
> > > [4] Pfeiffer, Jonas, et al. "AdapterFusion: Non-Destructive Task Composition for Transfer Learning." Proceedings of the 16th Conference of the European Chapter of the Association for Computational Linguistics: Main Volume. 2021.

---

### Author Response · Authors · 2021-11-23
**New Revision**

Dear Reviewers,

We have uploaded a new revision to our paper, in which we incorporate most of the comments from the reviewers. Specifically, we:

1. Added the comparison between LFPT5 and AdapterFusion in Section 4.4.
2. Added the ablation study on label consistency loss $\mathcal L^{\text{KL}}$ in Section 5.
3. Reported the state-of-the-art results for the original full-shot tasks in Section 4.2 and the accuracy of each dataset in Section 5.
4. Added experiments on learning from a large number of different domains (tasks) in Appendix A.3.
5. Added experiments on a different backbone model (T5-Base), different numbers of few-shot data, different number of pseudo samples and multiple prompts in multitask prompt tuning in Appendix A.4, A.5, A.6 and A.7, respectively.
6. Explained the number of tunable tokens per task prompt, few-shot settings and two regularization-based lifelong learning methods.
7. Replaced "vital" with "important".
8. Added suggested citations.
9. Fixed other minor comments (e.g., abbreviations)


We hope the reviewers have another look at the paper. We sincerely hope the reviewers acknowledge and respond to our responses so that we can address further concerns (if any) within the rebuttal time period.

We deeply appreciate your time and effort in reviewing our paper.

Best regards

---

### Decision · Program_Chairs · 2022-01-20

**Decision:**

Accept (Poster)

**Comment:**

This work defines the new problem of lifelong few-shot language learning where the goal is to continually learn new few-shot tasks and use those to benefit future tasks while not forgetting previous tasks. With larger models, this is an important goal due to the cost of updating and retraining these models. The work also shows superiority to existing approaches like EWC and MAS. After the author's rebuttal, the experimental section is also thorough with evaluation on a good range of tasks and approaches such as adapters showing good results. While this setting appears simpler than the full lifelong-learning setting and the approach combines existing ideas, this work's contribution to the definition and thinking about this problem is valuable. However, the authors should more clearly state the advantages of their approach vs standard prompt tuning (with an emphasis of benefiting future tasks) since two reviewers seem caught up on this point. The other two reviewers comments were addressed by the rebuttal as they stated in their comments.